# Glucagon-like peptide-1 receptor agonist in large vessel occlusion treated by reperfusion therapy—a phase 2 randomized trial

We aimed to determine the effect of semaglutide on patients with acute large vessel occlusion (LVO) receiving endovascular therapy (EVT). In this phase 2, investigator-initiated, multicenter, prospective, randomized, open-label, blinded endpoint trial conducted in China, we recruited patients with disabling LVO undergoing EVT. Patients were randomized to semaglutide therapy (0.5 mg subcutaneous semaglutide before and 1 week after EVT) or standard therapy. The primary outcome was defined as favorable neurological recovery (modified Rankin Scale 0–2 at 90 days). Between August 2023 and July 2024, 140 patients were randomized to semaglutide ($n = 69$) or standard therapy ($n = 71$). The primary outcome occurred in 39 (56.5%) in the semaglutide group and 39 (54.9%) in the standard therapy group (adjusted RR 1.05, 95% CI 0.95–1.15, $p = 0.37$). We observed treatment effect modification by intravenous thrombolysis (IVT) on semaglutide therapy ($p_{interaction} = 0.02$); thus we performed the following exploratory analyses: The primary outcome occurred in 22 (64.7%) in the semaglutide group and 15 (44.1%) in the standard therapy group (adjusted RR 1.18, 95% CI 1.02–1.36) in the no-IVT stratum ($n = 68$). The primary outcome was similar between two groups in the IVT-stratum. No severe adverse event was attributed to semaglutide treatment. This phase 2 trial suggested semaglutide was safe in patients with LVO and was associated with an improved neurological outcome in patients not receiving IVT. These preliminary observations should be confirmed in a phase 3 randomized trial (ClinicalTrials.gov Identifier: NCT05920889).

The glucagon-like peptide-1 (GLP-1) receptor is expressed in specialized neuronal subsets and glial cells widely distributed in the central nervous system[1], modulating a broad range of physiological functions from food intake, glucose homeostasis and metabolism[2–5], to systemic inflammation[6]. In rodent models of ischemic stroke, GLP-1 receptor agonists (GLP-1RAs) have been consistently shown to reduce infarct size, counteract reperfusion injury, and attenuate neuroinflammation[7–10]. These neuroprotective effects generalized across GLP-1RAs with variable blood-brain barrier (BBB) permeability (attributable to differences in properties such as molecular size, albumin binding[11], and potentially receptor-mediated transport into the brain)[12] and diverse experimental conditions[7], including various rodent species and strains, animal models with or without diabetes mellitus, stroke induction methods, treatment doses and regimens (initiating prior to, during, or even 1–3 days after stroke induction)[8,13], indicating a robust class effect.

Potential neuroprotective mechanisms of GLP-1RAs in acute ischemic stroke can be direct or indirect. Direct effects include attenuation of detrimental microglia activation[8], promotion of M2 microglia polarization[14], reduction of neuronal apoptosis and

✉e-mail: ho.ko@cuhk.edu.hk; che1971@126.com; ipyiuming@gmail.com

**Fig. 1 | CONSORT Flow Diagram.** This figure shows the overall patient flow in the trial. (BA basilar artery, CT computed tomography, IIT intention-to-treat, IV intravenous, NIHSS National Institutes of Health Stroke Scale).

promotion of neurogenesis[15], whereas indirect effects include amelioration of BBB breakdown[16], stabilization of blood glucose, modulation of systemic inflammation[6], and mimicking of ischemic preconditioning[17]. Collectively, these pleiotropic effects suggest that GLP-1RA use during the acute phase may offer neuroprotection in human ischemic stroke, aligning with the Stroke Treatment Academic Industry Roundtable X (STAIR X) recommendation for potential cytoprotective drugs[18].

To date, one phase 2 randomized trial demonstrated that exenatide, a GLP-1RA, did not improve early neurological recovery at 7 days post-stroke in a study population mainly consisting of mild strokes, despite a significant reduction of hyperglycemic events[19]. The potential neuroprotective effects of GLP-1RAs remain unexplored in patients with a higher severity of stroke undergoing reperfusion therapy. In large vessel occlusion (LVO) stroke patients with salvageable penumbra, successful reperfusion by endovascular therapy (EVT) may enable the therapeutic potential of GLP-1RAs to be manifested by: 1) enabling better delivery of the drug and its secondary mediators to vulnerable brain tissues with ischemic injury (even though this is yet to be verified in animal models comparing drug distribution with and without reperfusion under otherwise identical experimental conditions), 2) counteraction of reperfusion injury and neuroinflammation once blood flow is restored, and 3) stabilization of the BBB. Apart from enhancing the rescue and protection of ischemic penumbra, these neurovascular benefits could also synergize with the systemic anti-inflammatory and glucose-stabilizing properties of GLP-1RAs to improve stroke outcomes[20-22]. Therefore, investigating the use of a potent GLP-1RA in the context of LVO strokes reperfused by EVT may uncover the neuroprotective effects of GLP-1RAs in human subjects.

To inform the design of a phase 3 trial, the GALLOP (Glucagon-Like Peptide-1 Receptor Agonist in Large Vessel Occlusion Stroke Treated by Reperfusion Therapy) randomized trial aimed to determine the safety and signals for efficacy of semaglutide – currently recognized as the most clinically potent GLP-1RA[23]–in patients with acute LVO stroke treated by EVT.

## Results

### Trial population

The GALLOP trial is a phase 2, investigator-initiated, prospective, randomized, open-label, blinded endpoint trial conducted at two thrombectomy centers in China, which also serve as the secondary referral sites from ten primary stroke centers. We compared the efficacy and safety of semaglutide plus reperfusion therapy, i.e., EVT with or without intravenous thrombolysis (IVT), versus reperfusion therapy alone in patients with disabling LVO. Main eligibility criteria include: (1) LVO at the M1 segment of middle cerebral artery (MCA) or terminal internal carotid artery, (2) last-known-well (LKW) within 12 h at presentation, (3) Alberta Stroke Program Early Computed Tomography Score (ASPECTS) ≥ 6, and (4) for patients with a LKW between 6 and 12 h at presentation, computed tomography (CT) perfusion demonstrating a significant clinical-radiological or core-penumbra mismatch according to the DAWN or DEFUSE3 criteria[24,25], respectively (see Methods for details).

Between August 11, 2023, and July 25, 2024, 149 patients were screened for study eligibility (Fig. 1), among which 140 patients were enrolled and randomly assigned to semaglutide ($n = 69$) or standard therapy ($n = 71$). All patients randomized to the semaglutide group received semaglutide injection before puncture, while 3 (4.3%) patients in the semaglutide group did not complete day-7 semaglutide injection due to mortality. The modified Rankin Scale (mRS) at 90 days was missing for 3 (4.3%) patients in the semaglutide group and 2 (2.8%) patients in the standard therapy group due to loss to follow-up (Table S2). There was no other missing data. Minor protocol violations occurred in 3 (2.1%) patients (Table S3). The trial enrolled to

## Table 1 | Baseline patient characteristics

| | Semaglutide (n = 69) | Standard Therapy (n = 71) |
|---|---|---|
| Age mean ± sd | 69.2 ± 10.8 | 67.2 ± 11.1 |
| Male sex n (%) | 44 (63.8) | 50 (70.4) |
| Active smoker n (%) | 17 (24.6) | 16 (22.5) |
| Body weight (kg) mean ± sd | 70.3 ± 15.1 | 68.2 ± 12.5 |
| Hypertension n (%) | 42 (60.9) | 48 (67.6) |
| Diabetes mellitus n (%) | 10 (14.5) | 17 (23.9) |
| Atrial fibrillation n (%) | 21 (30.4) | 16 (22.5) |
| Ischemic heart disease n (%) | 11 (15.9) | 5 (7) |
| Congestive heart failure n (%) | 5 (7.2) | 1 (1.4) |
| History of stroke n (%) | 15 (21.7) | 8 (11.3) |
| Peripheral vascular disease n (%) | 1 (1.4) | 1 (1.4) |
| Chronic kidney disease n (%) | 0 (0) | 2 (2.8) |
| Secondary diversion n (%) | 8 (11.6) | 7 (9.9) |
| Systolic blood pressure mean ± sd | 145.8 ± 19.6 | 144.3 ± 23.5 |
| Diastolic blood pressure mean ± sd | 83.5 ± 11.7 | 85.1 ± 14.2 |
| Blood glucose on admission (mmol/L) mean ± sd | 7.1 ± 2.5 | 7.8 ± 3.2 |
| HbA1c (%) median (IQR) | 5.8 (5.5, 6.1) | 5.9 (5.5, 6.6) |
| ASPECTS median (IQR) | 8.5 (7, 10) | 8 (7, 9) |
| Infarct core (mL) median (IQR) | 9.1 (2.2, 32.3) | 13.2 (3.8, 25.2) |
| Premorbid mRS median (IQR) | 0 (0, 0) | 0 (0, 0) |
| Baseline NIHSS (IQR) | 16 (13, 20) | 16 (12, 20) |
| Intravenous thrombolysis n (%) | 35 (50.7) | 37 (52.1) |
| Collateral score median (IQR) | 1 (1, 2) | 2 (1, 2) |
| LKW-to-puncture (mins) median (IQR) | 323 (195, 455) | 336 (193, 427) |
| General anesthesia n (%) | 41 (59.4) | 52 (73.2) |
| mTICI 2c or above (%) | 54 (78.3) | 59 (83.1) |
| Aspiration n (%) | 32 (46.4) | 32 (45.1) |
| Stent retriever n (%) | 8 (11.6) | 6 (8.5) |
| Combined aspiration/stent retriever n (%) | 29 (42) | 33 (46.5) |
| Acute intracranial stenting n (%) | 21 (30.4) | 20 (28.2) |

*HbA1c* glycated hemoglobin A1c, *ASPECTS* Alberta Stroke Program Early Computed Tomography Score, *LKW* last-known-well, *mTICI* modified Thrombolysis in Cerebral Infarction score, *mRS* modified Rankin scale, *NIHSS* National Institutes of Health Stroke Scale.

completion. The mean age of patients was 68.2 ± 10.9 years, with 94 (67.1%) male and 46 (32.9%) female subjects. The median (interquartile range) National Institutes of Health Stroke Scale (NIHSS) was 16 (12, 20) and the baseline ASPECTS was 8 (7, 10). The median time (interquartile range) for onset to puncture was 333 (194, 440) minutes. Intravenous thrombolysis (IVT), either by alteplase 0.9 mg/kg or tenecteplase 0.25 mg/kg, was given to 35 (50.7%) in the semaglutide group and 37 (52.1%) in the standard therapy group. No patients received intraarterial thrombolysis. No patients had contrast extravasation on selective internal carotid artery angiography performed immediately after EVT. Baseline characteristics were balanced between the two groups (Table 1). All analyses were based on the intention-to-treat population, defined as all patients who were randomized (n = 140). Patients who were lost to follow-up were assigned with the worst

possible score for an outcome measure. Complete case analysis (n = 135) was also performed for patients with no loss to follow-up as a sensitivity analysis. There were no violations of the statistical assumptions of the modified Poisson regression models and proportional odds assumption for the ordinal logistic regression models. A pre-specified interim safety analysis conducted after the first 69 patients completed the study suggested no indication of increased risk of intracranial hemorrhage, malignant brain edema, neurological deterioration (as measured by changes in NIHSS from baseline to day 3), or poor neurological recovery (mRS 4–6 at 90 days) with semaglutide therapy (Table S4).

### Primary outcome

The primary outcome was efficacy, defined as achieving an mRS score of 0–2 at 90 days, a widely used scale to assess the degree of disability after stroke. This outcome occurred in 39 patients (56.5%) in the semaglutide group and 39 patients (54.9%) in the standard therapy group (adjusted risk ratio [RR] 1.05, 95% confidence interval [CI] 0.95–1.15, p = 0.37; Fig. 2A).

### Secondary outcomes

Secondary outcomes are shown in Figs. 2A, 3A and Table 2. The composite safety outcome, which comprised death, malignant brain edema (parenchymal hypodensity of at least 50% of the MCA territory with signs of local brain swelling such as sulcal effacement and compression of the lateral ventricle, and midline shift of ≥ 5 mm at the septum pellucidum or pineal gland with obliteration of the basal cisterns)[26] and intracranial hemorrhage (ICH, indicated by intracranial bleeding of Heidelberg Bleeding Classification of 2 or above)[27], occurred in 16 (23.2%) in the semaglutide group and 17 (23.9%) in the standard therapy group (adjusted RR 0.99, 95% CI 0.89–1.11). Semaglutide use was associated with a lower risk of ICH (5.8% vs. 15.5%, adjusted RR 0.91, 95% CI 0.83–0.99), and a higher likelihood of mRS 0–1 at 90 days (39.1% vs. 29.6%, adjusted RR 1.18, 95% CI 1.01–1.37). mRS 0–3 at 90 days, death, malignant brain edema and ordinal shift of mRS at 90 days were similar between the two arms. Final infarct sizes were not different between semaglutide and standard therapy (13.1 [4.9, 66.4] mL vs. 21.8 [10.5, 41.7] mL). Unadjusted risk ratios of primary and secondary outcomes are reported in Table S7.

### Exploratory analyses

We looked for treatment effect modification by covariates pre-specified in the statistical model. We found that IVT significantly interacted with the treatment effect by semaglutide ($p_{interaction}$ = 0.02). mTICI, and post-hoc analyses of diabetes mellitus and baseline blood glucose level did not have treatment effect modification with semaglutide.

Exploratory analyses by stratification according to IVT status were subsequently performed. In patients who did not receive IVT, the primary outcome occurred in 22 (64.7%) patients in the semaglutide group (n = 34) and 15 (44.1%) patients in the standard therapy group (n = 34) (adjusted RR 1.18, 95% CI 1.02–1.36, Fig. 2B). In patients who received IVT, the primary outcome occurred in 17 (48.6%) patients in the semaglutide group and 24 (64.9%) patients in the standard therapy group (adjusted RR 0.96, 95% CI 0.85–1.08, Fig. 2C). Baseline characteristics of both arms in each stratum were similar, except for a higher systolic blood pressure on presentation and lower collateral score among the semaglutide recipients in the IVT stratum (Tables S5A and S6A). The LKW-to-puncture time (median [interquartile range]) was shorter in the IVT stratum compared to the no-IVT stratum (262 [171, 397] minutes vs. 359 [248, 521] minutes, p = 0.002). The proportional odds ordinal logistic regression suggested treatment effect modification on the ordinal mRS shift by IVT (Fig. 3A–C). An ordinal shift towards better functional recovery was observed with

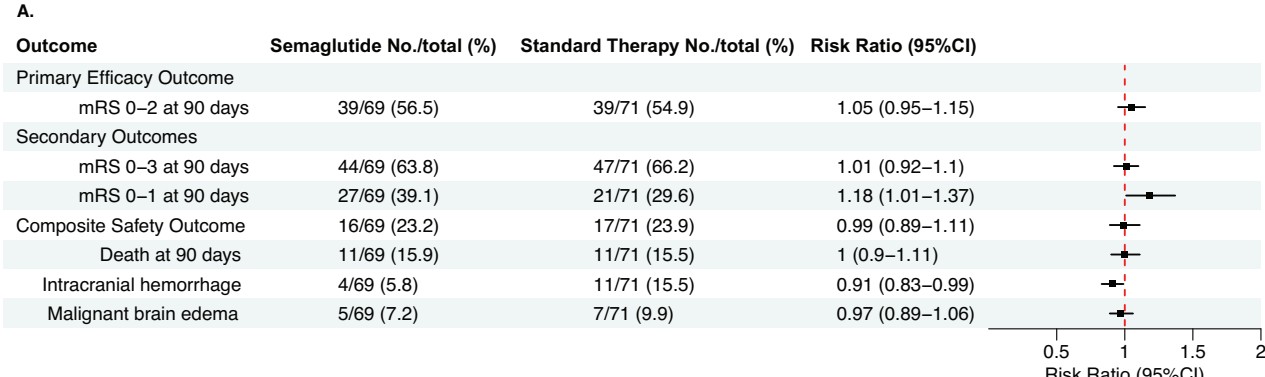

**Fig. 2 | Comparison of binary outcomes between semaglutide and standard therapy.** The event rates of the primary and binary secondary endpoints of semaglutide and standard therapy groups are shown in **A** the overall intention-to-treat sample (*n* = 140), **B** the no-IVT stratum (*n* = 68), and **C** the IVT stratum (*n* = 72). Data are presented as number (percentage) and risk ratios (95% CI). (CI confidence interval, IVT intravenous thrombolysis).

semaglutide therapy in the no-IVT stratum (adjusted odds ratio 2.92, 95% CI 1.21–7.2, Fig. 3B).

In the no-IVT stratum, we observed a higher proportion of mRS 0–1 at 90 days (44.1% vs. 26.5%, adjusted RR 1.18, 95% CI 1.02–1.37), mRS 0–3 at 90 days (73.5% vs. 55.9%, adjusted RR 1.14, 95% CI 1.00–1.30), greater reduction in day-3 NIHSS (−8 [−12, −4] vs. −3.5 [−6, −0.8]), and a lower day-3 blood glucose (6.4 ± 1.4 mmol/L vs. 7.8 ± 3.1 mmol/L) with semaglutide therapy (Table S5A–B). The change between day-3 and baseline blood glucose with semaglutide was modest (−0.4 ± 1.2 mmol/L vs. 0.2 ± 2.7 mmol/L). All secondary outcomes were not different between semaglutide and standard therapy in the IVT stratum (Figs. 2C, Fig. 3C and Table S6A–B).

Modified Poisson regression with restricted cubic spline found differing relationships between LKW-to-puncture time and risk ratios of achieving the primary outcome in the semaglutide and standard therapy arms (Fig. S1). In the standard therapy group, a longer LKW-to-puncture time was associated with a lower likelihood of the primary efficacy outcome, whereas this association was diminished in the semaglutide group.

### Adverse events and sensitivity analysis

The rates of adverse events are listed in Table 3. Death occurred in 11 (15.9%) patients in the semaglutide group and 11 (15.4%) patients in the standard therapy group (unadjusted RR 1.00, 95% CI 0.9–1.11). Symptomatic ICH, defined as ICH with NIHSS increase of ≥ 4 points, occurred in 1 (1.4%) patient in the semaglutide group and 7 (9.9%) patients in the standard therapy group (unadjusted RR 0.92, 95%CI 0.85–0.98). No patients developed hypoglycemia. Complete case analysis showed similar findings (Tables S8–10).

## Discussion

In this phase 2, investigator-initiated, prospective, randomized, open-label, blinded endpoint trial that investigated the use of semaglutide therapy in patients with LVO strokes treated by EVT, semaglutide was

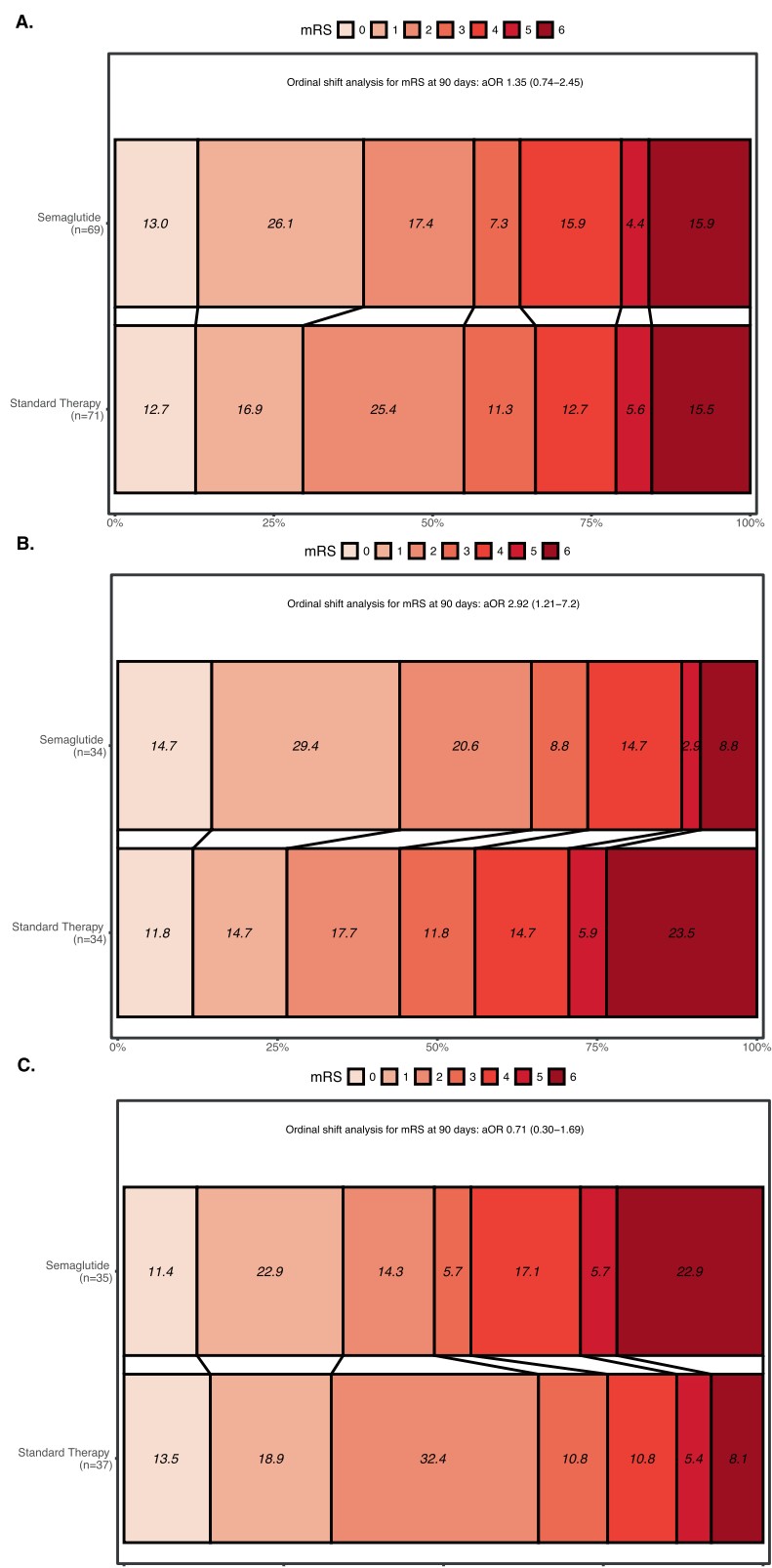

**Fig. 3 | Distribution of functional outcomes at 90 days by treatment groups.** mRS scores are shown in **A** the overall intention-to-treat sample (*n* = 140), **B** the no-IVT stratum (*n* = 68), and **C** the IVT stratum (*n* = 72). Scores range from 0 to 6: 0 = no symptoms, 1 = symptoms without clinically significant disability, 2 = slight disability, 3 = moderate disability, 4 = moderately severe disability, 5 = severe disability, and 6 = death. (aOR adjusted odds ratio, CI confidence interval, IVT intravenous thrombolysis, mRS modified Rankin Scale).

**Table 2 | Secondary continuous outcomes of the intention-to-treat population**

| Secondary outcomes | Semaglutide (n = 69) | Standard Therapy (n = 71) |
|---|---|---|
| Change between baseline and D3 NIHSS median(IQR) | −8 (−12, −2) | −5 (−9.5, −1.5) |
| Final infarct size (mL) median(IQR) | 13.1 (4.9, 66.4) | 21.8 (10.5, 41.7) |
| Blood glucose on day 3 (mmol/L) mean ± sd | 7 ± 2.6 | 7.9 ± 3.1 |
| Change between baseline and D3 glucose (mmol/L) mean ± sd | 0.2 ± 1.6 | 0.1 ± 3 |

NIHSS National Institutes of Health Stroke Scale.

**Table 3 | Adverse events**

| | Semaglutide (n = 69) | Standard Therapy (n = 71) |
|---|---|---|
| Death n (%) | 11 (15.9) | 11 (15.4) |
| ICH n (%) | 4 (5.8) | 11 (15.4) |
| Symptomatic ICH n (%) | 1 (1.4) | 7 (9.9) |
| Malignant brain edema n (%) | 5 (7.2) | 7 (9.9) |
| Pneumonia n (%) | 4 (5.8) | 2 (2.8) |
| Nausea/Vomiting n (%) | 3 (4.3) | 1 (1.4) |
| Hypoglycemia n (%) | 0 (0) | 0 (0) |
| Recurrent ischemic stroke n (%) | 1 (1.4) | 0 (0) |
| Infection (excluding pneumonia) n (%) | 3 (4.3) | 1 (1.4) |
| Hypotension n (%) | 0 (0) | 1 (1.4) |
| Euglycemic diabetic ketoacidosis n (%) | 1 (1.4) | 1 (1.4) |
| Allergic reactions n (%) | 0 (0) | 0 (0) |

N.B. Intracranial hemorrhage (ICH) is defined as Heidelberg bleeding classification class 2 or above. Symptomatic ICH is defined as ICH with an increase in National Institute of Health Stroke Scale ≥4.
ICH intracranial hemorrhage.

well-tolerated without signals of harm. Semaglutide treatment did not improve functional recovery in the overall population but was associated with a reduced risk of ICH. Furthermore, semaglutide treatment was associated with a better functional outcome at 90 days in patients who had not received IVT. These findings require confirmation in a larger trial.

Preclinical neuroprotective effects of GLP-1RAs were demonstrated in unilateral transient MCA occlusion animal models, showing reduced infarct volume, oxidative stress, neuroinflammation, neuronal loss and neurological deficit in both diabetic and non-diabetic rodents[7]. Notably, the potential neuroprotective effects of GLP-1RA were evident even with delayed drug administration. In normoglycemic mice, a two-week course of daily liraglutide initiated 24 h after stroke induction reduced infarct size, improved neurological recovery, and promoted angiogenesis[13]. In another study involving diabetic mice, delayed administration of GLP-1RA 3 days after stroke induction for a duration of 8 weeks led to improved motor recovery and normalization of micro-vessel density and pericyte coverage[8]. Preclinical evidence also suggests that GLP-1RA may enhance BBB integrity, which might have reduced ischemia- or reperfusion-related brain hemorrhage following EVT as observed in our study[28,29]. Systemically, GLP-1RA may reduce post-stroke stress hyperglycemia, which is a risk factor of BBB dysfunction and poor clinical outcomes[20].

Clinical evidence exploring the use of GLP-1RA in acute ischemic stroke is scarce. Thus far, only the phase 2 TEXAIS (Treatment with Exenatide in Acute Ischemic Stroke) randomized trial has been conducted, with the primary outcome defined as NIHSS improvement ≥ 8 (or NIHSS scores 0−1) at 7 days post-stroke. Focusing on this outcome measure, TEXAIS did not demonstrate benefit with exenatide in

patients with acute ischemic stroke within 9 h of onset, despite reporting a reduction in hyperglycemic events with exenatide[19]. The GALLOP trial differs from the TEXAIS in several aspects. First, we included only patients with severe ischemic stroke. This excluded mild stroke with a high likelihood of achieving excellent outcomes, which may offer only a small therapeutic window for the potential treatment effects of GLP-1RA to manifest. Second, we focused on patients eligible for EVT. The high rates of excellent angiographic outcomes may have created conditions more similar to those in transient MCA occlusion animal models. Thirdly, we chose semaglutide, which may be more efficacious than exenatide in terms of cardiovascular protection and regulation of systemic stress and inflammatory responses[30,31], as reflected in post-stroke glucose stabilization.

The treatment effect modification observed with IVT on semaglutide was unexpected. At present, there are no available preclinical or clinical studies that evaluated potential interactions between semaglutide and tenecteplase or alteplase[32]. Any potential pharmacokinetic interaction between GLP-1RA and tissue plasminogen activators should be excluded. On the other hand, GLP-1RA may inhibit the expression of plasminogen activator inhibitor-1 (PAI-1)[33,34], a molecule that inhibits tissue plasminogen activators and the conversion of plasminogen to plasmin[35]. Downregulation of PAI-1 with co-administration of IVT may thus indirectly potentiate plasmin generation and undesired effects of thrombolytic agents, such as BBB disruption[36], brain edema[36], and neurotoxicity[37,38]. Nonetheless, these postulations require further confirmation. Importantly, the interaction between semaglutide and IVT could be confounded by time metrics, as IVT administration is a proxy of early presentation. As both neuroinflammation and reperfusion injury are time-dependent in ischemic stroke, the therapeutic benefits of GLP-1RAs in modulating or counteracting these processes may only become more apparent beyond the conventional 4.5-h time window for IVT administration, within which the maximum treatment efficacy is primarily determined by reperfusion therapy. Owing to the limited sample size inherited to the phase 2 trial design and the exploratory nature of this analysis, the possibility of type I error cannot be excluded. Thus, an adequately powered phase 3 study is needed to explore whether the potential neuroprotective effect of GLP-1RA is more pronounced in strokes with later presentations.

Our study has several limitations. First, the current phase 2 trial was not powered to draw definite conclusions regarding the efficacy of semaglutide in improving neurological outcomes in patients with LVO, but rather to estimate any potential treatment effects and evaluate the safety and tolerability of semaglutide treatment to inform the design for a large phase 3 trial. Second, as randomization only took place in two thrombectomy centers, future studies should consider evaluating GLP-1RA administration at primary stroke centers prior to secondary transfer for EVT. Expanding the number of participating centers and accounting for heterogeneity across sites are also necessary to provide a more accurate estimate of the treatment effect and enhance the generalizability of the findings. Third, whether similar observations could be reproduced with different dosages of semaglutide require further study. Fourth, the effect of semaglutide in LVO patients across different ages, LKW-to-presentation time, body weight, ASPECTS, collateral status, blood

glucose level on presentation requires further analyses. Fifth, the original plan to assess BBB permeability as a radiological outcome and to conduct omics analyses was not implemented due to the lack of model validation and resource limitations, respectively. Future studies should incorporate BBB leakage-associated biomarkers (e.g., matrix metalloproteinase-9) to more effectively evaluate the impact of GLP-1RA on the BBB[39]. Last, placebo was not available, but we ensured that all operating neuro-interventionists and raters of functional outcomes were blinded from the knowledge of the treatment allocation. All radiological parameters were determined by the central core laboratory with human raters or computer algorithms blinded from treatment allocation and functional outcomes.

In conclusion, 0.5 mg semaglutide administration before and 1 week after EVT in patients with LVO onset within 12 h was safe and well-tolerated. Semaglutide treatment did not improve functional recovery in the overall population but was associated with lower risk of intracranial hemorrhage, and an improved neurological outcome in patients who did not receive IVT. These preliminary observations should be confirmed in a large phase 3 trial.

## Methods

### Study design and participants
The GALLOP trial is a phase 2, investigator-initiated, prospective, randomized, open-label, blinded endpoint (PROBE) trial conducted at two thrombectomy centers in China, which also serve as secondary referral sites from ten other primary stroke centers. We compared the efficacy and safety of semaglutide plus reperfusion therapy, i.e., EVT with or without intravenous thrombolysis (IVT), versus reperfusion therapy alone in patients with disabling LVO. Key inclusion criteria were adults age 18 years or older, LVO at the terminal internal carotid artery or M1 segment of the MCA, ASPECTS 6 to 10, NIHSS ≥ 10 at randomization, stroke onset or LKW ≤ 12 hours at randomization, and pre-stroke mRS ≤ 2. Patients who presented between 6 and 12 h from symptom onset or LKW required a CT perfusion scan to assess for clinical-core or perfusion-core mismatch according to the DAWN and DEFUSE3 criteria[24,25]. The DAWN criteria required a mismatch between clinical deficit and infarct core volume as determined by CT perfusion: patients aged ≥ 80 years needed an NIHSS score ≥ 10 and infarct core (volume of relative cerebral blood flow [rCBF] < 30%) < 21 mL; those < 80 years needed NIHSS ≥ 10 and core < 31 mL, or NIHSS ≥ 20 and core 31 – < 51 mL. The DEFUSE3 criteria required imaging evidence of a target mismatch profile: infarct core (volume of rCBF < 30%) < 70 mL, penumbral tissue (volume of Tmax > 6 s - volume of rCBF < 30%) ≥ 15 mL, and a mismatch ratio (volume of Tmax > 6 s: volume of rCBF < 30%) of ≥ 1.8. Full inclusion and exclusion criteria are enlisted in Table S1. The trial was registered with ClinicalTrials.gov (NCT05920889) and approved by the local institutional review boards (Joint CUHK-NTEC Clinical Research Ethics Committee Reference No.: 2023.026, Science Research Ethics Committee Linyi People's Hospital Reference No.: YX200651). As the trial did not involve genetic information and materials, it was waived approval from China's Ministry of Science and Technology related to the export of genetic information and materials. A detailed study protocol and statistical analysis plan can be found in the Study Protocol. The study protocol and statistical analysis plan had been revised once after commencement of study on August 1 2023 (see Supplementary Note 1, Study Endpoints and Statistical Analysis).

### Randomization and masking
All potentially eligible patients underwent computed tomography angiography (CTA) to confirm the LVO. We obtained written informed consent from patients or their legal representatives upon confirmation of study eligibility by two investigators. Patients were then randomized in a 1:1 ratio to receive semaglutide plus EVT or EVT alone in the emergency department. Permuted blocked

randomization was employed in the study to ensure the balance of subjects throughout the trial setting[40]. Randomly generated block sizes of 2, 4, and 6 were adopted to avoid possible mid-block inequality caused by larger blocks. Two blocks with unbalanced treatment distribution were generated at the start and middle of the list. The randomization process was performed using the blockrand package (v1.5) in R studio (v4.4.1, R Project for Statistical Computing, RStudio Team 2022). Treatment allocation was concealed until study eligibility was confirmed by two investigators and a written informed consent had been obtained.

### Procedures
All consecutive patients with LVO who planned to undergo EVT were screened for study eligibility. In addition, study participants underwent screening for intravenous thrombolysis (IVT) by alteplase or tenecteplase before the randomization process. In general, patients with an LKW-to-presentation time of ≤ 4.5 h were considered for IVT unless otherwise contraindicated[41]. Patients in the semaglutide arm received subcutaneous semaglutide 0.5 mg before arterial puncture and 7 days after EVT. Semaglutide administration was documented using the computerized systems at the participating centers, either in the stroke unit or rehabilitation unit. The regimen was chosen based on observational studies that suggested an increase in early BBB permeability was associated with worse functional outcomes[42], while a subacute BBB permeability increase beyond 10 days after ischemic stroke was associated with good functional outcomes[43]. Operating neurointerventionalists were masked from the treatment allocation. EVT devices were deployed at the discretion of treating interventionists. All study participants were transferred to the stroke unit or neuro-intensive care unit after the procedure, where they received stroke rehabilitation and secondary stroke prevention according to the Chinese national guidelines[44].

We prospectively collected patients' demographic characteristics, medical history, laboratory results and stroke-related parameters including the NIHSS on admission, LKW-to-puncture time and periprocedural details (mode of anesthesia, procedure time, modified Thrombolysis in Cerebral Infarction [mTICI] score). All patients received a follow-up NIHSS reassessment three days after the procedure. Plain CT brain was repeated 4 to 7 days or during clinical deterioration to detect any intracranial hemorrhage (ICH) or malignant brain edema (MBE). Brain magnetic resonance imaging (MRI) was obtained 14 to 21 days of randomization for quantification of final infarct size. Clinical follow-up was arranged 90 days after randomization for evaluation of functional recovery by mRS. A telephone or video conferencing follow-up was arranged for patients who were unable to attend an in-person follow-up. All certified assessors for NIHSS and mRS were blinded from the treatment allocation and were not the operating neuro-interventionists of the study subjects.

### Radiological parameters
Images were processed in the central core image processing laboratory. ASPECTS, volumes of infarct core and total ischemic territory on CT perfusion, and final infarct size on MRI brain were determined by in-house automated image-processing algorithms validated with commercially available software. Collateral score[45], mTICI score[46], and hemorrhagic complications were determined by two raters (BI, JA) with more than 10 years of experience[27]. Disparities were resolved by a third rater (SM). Raters and computer algorithms were blinded from treatment allocation and clinical outcomes.

### Study endpoints
The primary outcome was efficacy, defined as mRS of 0 to 2 at 90 days. The original definition of mRS of 0 to 3 at 90 days was not used after the steering committee discussion on 15 April 2024 as most clinical trials adopted mRS 0 to 2 as the definition for good

neurological recovery for anterior circulation LVO (Supplementary Note 1)[47,48]. In the study protocol, a primary composite safety endpoint comprising death, MBE, and ICH was defined. This endpoint was analyzed as a secondary outcome, as no statistical power had been prespecified for its assessment. Other secondary outcomes were the ordinal shift in mRS at 90 days, mRS 0 to 3 at 90 days, mRS 0 to 1 at 90 days, death, final infarct size on MRI 14–21 days after randomization, MBE (parenchymal hypodensity of at least 50% of the MCA territory with signs of local brain swelling such as sulcal effacement and compression of the lateral ventricle, and midline shift of ≥ 5 mm at the septum pellucidum or pineal gland with obliteration of the basal cisterns)[26], and ICH (hemorrhage occupying more than 30% of the infarcted tissue with mass effect, and/or hemorrhage outside the infarcted brain tissue, or any intracerebral-extracerebral hemorrhage, such as hematoma in parenchymal tissue remote from the infarcted area, intraventricular hemorrhage, subarachnoid hemorrhage, or subdural hemorrhage)[27]. Secondary exploratory outcomes included changes between day-3 and baseline NIHSS and day-3 and baseline blood glucose level.

### Sample size estimation
No human trials had evaluated the efficacy and safety of GLP-1RA on EVT-eligible patients. We hypothesized that semaglutide had a mild-to-moderate effect size, corresponding to a standardized difference of 0.15–0.25, in achieving good functional outcome, i.e., the primary efficacy outcome of mRS 0–2 at 90 days, in patients eligible for the study. Considering a rate of 2.5% suboptimal scan qualities, 5% of loss-to-follow-up and 10% of suboptimal recanalization according to the track records of participating centers, 140 participants were required for 90% power and significance level of 0.05 for the main trial[49]. Details of the sample size estimation are described in the Study Protocol.

### Statistical analyses
We expressed normally distributed continuous variables as mean ± standard deviation and non-normally distributed continuous variables as median (interquartile range [IQR]). Categorical variables were expressed as number (percentage). We compared baseline continuous variables of semaglutide versus standard therapy by independent sample *t*-test or Wilcoxon rank-sum test, and categorical variables by Chi-squared or Fisher's exact test as appropriate.

We used the modified Poisson regression models (Poisson regression with a robust error variance estimator to correct standard errors for binary data) to evaluate the risk ratios (RR) of the binary primary and secondary outcomes of semaglutide versus standard therapy[50]. The original plan of multivariable logistic regression was not used as modified Poisson regression could provide a more unbiased estimation of risk ratios in the setting of common event occurrence compared to logistic regression (Supplementary Note 1)[51]. The goodness-of-fit of the modified Poisson regression models was assessed using the normalized sum of square test[52]. Proportional odds ordinal logistic regression was used to compare the ordinal shift of mRS between the two groups. The proportional odds assumption was tested with the Brant test[53]. Cox regression was used to compare mortality at 90 days. Analysis of covariance (ANCOVA) was used to compare continuous outcomes. Comparisons of the primary efficacy outcome, safety outcome, ordinal shift of mRS, mRS 0–3 and mRS 0–1 at 90 days were adjusted for a pre-specified set of covariates, including age, premorbid mRS, IVT status, NIHSS on presentation, LKW-to-puncture time, mTICI, and baseline ASPECTS. Comparisons of other secondary outcomes were adjusted for NIHSS on presentation. All analyses were performed for the intention-to-treat population, defined as patients who underwent randomization, regardless of treatment received. Deceased patients were considered having an NIHSS of 42. Missing outcomes were imputed as the worst

possible score for an outcome measure, i.e., an mRS of 6 and an NIHSS of 42. Complete case analyses for the outcomes were performed in patients without missing data. We performed post-hoc analyses by stratifying IVT status due to the evidence of treatment effect modification by IVT treatment on semaglutide therapy. Modified Poisson regression with restricted cubic spline was used to depict the relationship between LKW-to-puncture and risk ratios of achieving the primary outcome. All secondary outcomes and post-hoc analyses should be considered exploratory. All statistical analyses were performed using R studio (v4.4.1, R Project for Statistical Computing, RStudio Team 2022).

### Reporting summary
Further information on research design is available in the Nature Portfolio Reporting Summary linked to this article.

## Data availability

The data generated in the figures have been deposited in the Figshare database (https://doi.org/10.6084/m9.figshare.28089197). Anonymized data, including age, sex, admission NIHSS, premorbid mRS, treatment allocation, and study outcomes will be made available by requesting the corresponding authors (Bonaventure Y. Ip, email: ipyiuming@gmail.com, or Ho Ko, email: ho.ko@cuhk.edu.hk) from qualified investigators for academic purposes, beginning 3 months and ending 5 years following publication. The corresponding authors will reply to the request within 2 months, subject to the approval of the Joint Chinese University of Hong Kong-New Territory East Cluster and the Linyi People's Hospital Clinical Research Ethics Committees. Source data are provided with this paper.

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

## Acknowledgements

Non-author contributions to Data collection, Analysis, or Writing/editing assistance: The authors thank the contributions from Dr. Robert Dan (MBBS, The Chinese University of Hong Kong), Mrs. Sophia Dan (BBS, MH, JP, The Chinese University of Hong Kong), Ms. Anki Miu (MSc, The Chinese University of Hong Kong), Mr. Ka Hung Li (MSc, Hong Kong Baptist University) and Ms. Maggie Lau (MSc, The Chinese University of Hong Kong). Dr. Robert Dan provided general advice for the study. Mrs. Sophia Dan provided general advice for the study. Ms. Anki Miu provided clerical assistance for the study. Mr. Ka Hung Li provided general advice for the study. Ms. Maggie Lau provided general advice for the study. Access to data and data analysis: Bonaventure Y. Ip and Ho Ko have full access to all the data in the study and takes responsibility for the integrity of the data and the accuracy of the data analysis.

## Author contributions

H.W. conducted the study, collected the data, and critically revised the manuscript for intellectual content. T.W.L., J.H., J.S., Y.Liang, J.Z., Q.C., W.Z., Y.Li, S.H.M., W.T.L., J.C., C.Chan, J.W., A.J.K., K.M., F.F., A.C., V.I., H.Leung., Y.S., K.T.W., B.Lai, C.Chu, H.S.L., A.H., T.C., J.A., S.H.L., L.C., J.Y., L.L., T.H. and S.F.T. conducted the study and collected the data. H.Li. verified the data. S.P. and T.Y. conducted statistical analysis. X.L., B.Lam, V.C.T.M., R.H.M.C., T.N.N., W.H., F.C. critically revised the manuscript for intellectual content. H.K. and B.Y.I. created the study concept and design, supervised the study, conducted statistical analysis, verified the data, wrote and critically revised the manuscript for intellectual content.

## Competing interests

T.N.N. reports Associate Editorship of Stroke; Advisory boards of Brainomix and Aruna Bio; Speaker for Genentech and Kaneka; Consulting for Medtronic. All other authors report no disclosures relevant to the manuscript.

## Additional information

Hao Wang[1,14], Ho Ko[2,3,4,14]✉, Thomas W. Leung[2,3,4,5,14], Junzhe Huang[2,3,4], Junjie Sai[1], Yu Liang[1], Haipeng Li[2], Jie Zhang[1], Qingyang Cao[1], Wentao Zang[1], Yinfei Li[1], Sze Ho Ma[2], Wai Ting Lui[2], Joseph Choi[2], Charlie Chan[2], Jason Wong[2], Andrew J. Kwok[2,3,4], Karen Ma[2], Florence Fan[2], Anne Chan[2], Vincent Ip[2], Howan Leung[2], Yannie Soo[2], Ka Tak Wong[6], Billy Lai[6], CM Chu[6], Ho Sang Leung[6], Anselm Hui[6], Tom Cheung[6], Jill Abrigo[6], Siu Hung Li[7], Larry Chan[8], Jonas Yeung[8], Sangqi Pan[2], Terry Yip[2,3,9], LT Lui[10], Trista Hung[2], Suk Fung Tsang[2], Xinyi Leng[2], Bonnie Lam[2], Vincent C. T. Mok[2,3,4], Rosa H. M. Chan[10,11], Thanh N. Nguyen[12], Wei Hu[13,14], Fengyuan Che[1]✉ & Bonaventure Y. Ip[2,3,4,5]✉

[1]Department of Neurology, Linyi People's Hospital, Affiliated Hospital of Shandong Second Medical University, Linyi, China. [2]Division of Neurology, Department of Medicine and Therapeutics, Faculty of Medicine, The Chinese University of Hong Kong, Hong Kong SAR, China. [3]Li Ka Shing Institute of Health Sciences, Faculty of Medicine, The Chinese University of Hong Kong, Hong Kong SAR, China. [4]Gerald Choa Neuroscience Institute, The Chinese University of Hong Kong, Hong Kong SAR, China. [5]Kwok Tak Seng Centre for Stroke Research and Intervention, The Chinese University of Hong Kong, Hong Kong SAR, China. [6]Department of Imaging and Interventional Radiology, Faculty of Medicine, The Chinese University of Hong Kong, Hong Kong SAR, China. [7]Department of Medicine, North District Hospital, Hong Kong SAR, China. [8]Department of Medicine, Alice Ho Miu Ling Nethersole Hospital, Hong Kong SAR, China. [9]Medical Data Analytic Center, Faculty of Medicine, The Chinese University of Hong Kong, Hong Kong SAR, China. [10]Hong Kong Centre for Cerebro-Cardiovascular Health Engineering, Hong Kong SAR, China. [11]Department of Electrical Engineering, City University of Hong Kong, Hong Kong SAR, China. [12]Department of Neurology, Neurosurgery, and Radiology, Boston Medical Center, Boston University Chobanian and Avedisian School of Medicine, Boston, MA, USA. [13]Department of Neurology, The First Affiliated Hospital of USTC, Division of Life Sciences and Medicine, University of Science and Technology of China, Hefei, China. [14]These authors contributed equally: Hao Wang, Ho Ko, Thomas W. Leung, Wei Hu. ✉e-mail: ho.ko@cuhk.edu.hk; che1971@126.com; ipyiuming@gmail.com

