## [Transparent Peer Review file · Nature Communications]

Glucagon-Like Peptide-1 Receptor Agonist in Large Vessel Occlusion Treated by Reperfusion Therapy – A Phase 2 Randomized Trial

Corresponding Author: Dr Bonaventure Ip

Version 1:

Reviewer comments:

Reviewer #3

(Remarks to the Author)

The work by Hao Wang and colleagues attempts addressing the safety and potential efficacy of GLP-1RA Semaglutide to improve neurological recovery in Large Vessel Occlusion stroke patients Treated by Reperfusion Therapy – in a Phase 2 Randomized Trial. This, in light of a possible Phase 3 clinical trial.

The work is timely and very relevant after a vast number of successful preclinical studies showing neuroprotection and improved functional recovery by GLP-1R agonists. However, some points need to be clarified to improve the manuscript.

1) The Introduction does not clearly motivate and explain the rationale of the study.

a. References 1-2 should be replaced with specific references related to GLP-1R expression in the brain (the original works). Moreover, it should be mentioned that the permeability of the brain to these drugs is uncertain (and provide references) and that CNS effects could be obtained indirectly, either to modulate normal physiology or in pathological conditions such as stroke.

b. References 8-10 should be replaced/completed with the first original works showing the neuroprotective efficacy of GLP-1RA in animal models of stroke.

i. Li et al Proc Natl Acad Sci U S A. 2009 (in naïve rodents)

ii. Darsalia et al Cli Sci 2012 (in diabetic rodents)

iii. Moreover, the first and only available work showing neurological efficacy of GLP-1RA when given post-acute (from Day 3 after stroke) was provided by Augestad et al Br J Pharmacol 2012 and this study may be cited since strictly relevant for the ms.

c. The reference 12 should be commented in more detail: I do not think that this study was designed to assess the neuroprotective efficacy/improve clinical outcome.

d. The reference 13 should be checked since I am unsure if it supports the statement made.

e. The text end of Intro only considers possible direct effect of Semaglutide in the brain after EVT. However, as commented above, the bbb may not allow the passage of GLP-1RA and thus alternative indirect ways may be added to the hypothesis of the work.

2) Results section

a. The Trial Population section in Results should be improved with a clear Study Design to complement Figure 1 or to add to Figure 1. It should be clearer the trial flow with also IVT treated patients.

b. It is well known that confounding factors for GLP-1RA treatment are glucose, blood pressure and BMI. The authors should expand on whether and how these factors have been handled in the study in both Results and Discussion

3) Discussion

a. In relation to reference 14, see comment above when in the study by Augestad et al GLP-1RA was given from day 3 post-stroke

b. Pls clarify better the following statement in Discussion page 9 “On the one hand, any potential direct interference of GLP-1RA activity by tenecteplase or alteplase, or vice versa, should be ruled out”.

c. Acknowledge the fact of the limited number of patients and whether, if known, the results will be sufficient to trigger a phase 3.

d. In the results section the authors state: “The onset to puncture time was shorter in the IVT stratum compared to the no-IVT stratum (262 [171-397] minutes vs. 359 [248-521] minutes, $p=0.002$)” – an almost 2-hour difference. Timing of intervention is the most important predictor of neuroprotective efficacy. This should be discussed. Is it possible that in IVT strata, the maximum effect on efficacy outcome was already achieved by timely reperfusion and thus the modifying effect of Semaglutide was not detectable? – this should be discussed. The authors should also go deeper and analyze the effect of intervention time on efficacy outcome.

Reviewer #4

(Remarks to the Author)

This study explores the innovative application of a Glucagon-Like Peptide-1 Receptor Agonist as an adjunctive neuroprotective therapy in patients undergoing reperfusion treatment, a topic of significant clinical interest. However, the manuscript’s scientific rigor is compromised by critical ambiguities in study design and endpoint definitions. A central issue lies in the unclear designation of the primary endpoint. The authors simultaneously emphasize both safety (e.g., symptomatic intracerebral hemorrhage) and efficacy (e.g., functional independence) as dual primary endpoints without hierarchical prioritization or statistical adjustments for multiplicity. This dual focus creates interpretative challenges: if efficacy is the primary objective, the absence of foundational Phase I data—such as pharmacokinetics (e.g., blood-brain barrier penetration), dose-escalation studies, or biomarker evidence of neuroprotection—undermines the rationale for directly advancing to a Phase II trial. Conversely, if safety is prioritized, the sample size justification based on efficacy outcomes (e.g., functional independence rates) becomes methodologically inconsistent, as safety endpoints typically require distinct statistical assumptions (e.g., non-inferiority margins for adverse events versus superiority for clinical outcomes). To strengthen the manuscript, the authors must clarify the study’s phase and primary objective.

I have the following comments which the authors should address

1.
Line 52 The primary endpoint should be clearly specified here, identifying which outcome serves as the main endpoint.
- 2..
71 The authors claim that the drug aligns with STAIR criteria; however, they have not sufficiently demonstrated the relevant preclinical research progress, such as dose-response relationships, studies in animals of different sexes, different model or data from aged animal models.
3.
83 84 Does research evidence exist to support the claims that reperfusion improves the distribution of this drug and enhances its anti-inflammatory effects?
4.
96 In how many centers was .the study conducted?
- 5
110 All patients who received randomized allocation?
6.
111 It is challenging to draw confirmatory efficacy conclusions from this Phase II study.
7.
119 This discrepancy is atypical. How did the authors define and adjudicate the occurrence of ICH (intracerebral hemorrhage)?
8.
156 A symptomatic intracerebral hemorrhage (SICH) rate of 9.9% was observed in the control
9.
228 group, which is unusually high. What factors might explain this elevated incidence?
Heterogeneity across participating centers should be acknowledged as a limitation of this study
10.
237 What were the specific inclusion criteria for hemodynamic parameters (e.g., cerebral blood flow [CBF], time-to-peak [TTP], mean transit time [MTT]) derived from perfusion imaging in this study?

11.
293 The endpoint configuration here is confusing—is safety or efficacy designated as the primary endpoint? This ambiguity must be clarified to ensure methodological rigor.

12.
307 The sample size justification is similarly unclear and lacks empirical foundation. If, as the authors state, no prior human studies have investigated this drug in clinical settings, it would be methodologically imperative to first conduct a first-in-human (FIH) Phase I trial to obtain critical safety data (e.g., dose-escalation findings), pharmacokinetic/pharmacodynamic (PK/PD) parameters, and preliminary evidence of biological activity. These data are essential for rationally estimating the sample size required for subsequent efficacy-focused studies (e.g., Phase II). The authors must provide further clarification on this process.

13.
315 The statistical model employed for the primary endpoint must be thoroughly elaborated, including its rationale, assumptions, and adjustments. This is critical to ensure the validity and reproducibility of the findings. Key details—such as covariate adjustment, handling of missing data, and validation of model assumptions (e.g., normality, proportional hazards)—are omitted, undermining the credibility of the results.

Specific Recommendations:

Explicitly state whether the analysis used a generalized linear model (GLM), Cox proportional hazards model, mixed-effects model, or another approach. Provide the mathematical formulation if possible

Clarify if and how baseline covariates (e.g., age, baseline severity) were included. Were interactions tested? How were violations of model assumptions (e.g., non-linearity in logistic regression) addressed?

Describe sensitivity analyses (e.g., alternative models, different imputation methods; per-protocol analysis) to confirm the robustness of the primary endpoint results.

Are all analyses based on the ITT population (intention-to-treat), as defined in the protocol?

Was a safety set analysis omitted, and if so, justify its exclusion despite being standard in safety reporting.

Are Multiplicity adjustments used in this analysis? Methods to control type I error inflation from multiple hypothesis testing.

Reviewer #5

(Remarks to the Author)

Reviewer #6

(Remarks to the Author)

This phase 2 randomized trial evaluated the safety and efficacy of semaglutide, a GLP-1 receptor agonist, in 140 Chinese patients with acute anterior circulation large vessel occlusion stroke treated with endovascular therapy (EVT). Semaglutide did not improve the primary efficacy endpoints, functional independence at 90 days (mRS 0–2) and a composite safety outcome, compared to standard care, but it was significantly associated with lower rates of symptomatic ICH and better neurological recovery. An exploratory subgroup analysis suggested benefit in patients not receiving IV thrombolysis, though this was not pre-specified and is subject to inflated type I error. While the trial was well conducted and clearly reported, the clinical significance is limited given the null primary outcome, lack of adjustment for multiple comparisons, and post hoc changes. The findings can be viewed as hypothesis-generating and require validation in phase 3 trials before influencing practice. The manuscript is largely consistent with the statistical analysis plan (SAP) in its study design, eligibility criteria, intervention protocol, outcome definitions, and statistical methods. However, several inconsistencies and omissions should be addressed to align fully with the SAP and registrations and ensure transparency. My specific comments are as follows:

1. In the 2025/03/04 trial registration update, the PI revised several outcome measures, including the addition of a second primary endpoint: a composite safety outcome comprising death, intracranial hemorrhage (ICH), and malignant brain edema (MBE). Several secondary outcomes were newly specified or clarified, including mRS 0–1 and 0–3 at 90 days, mortality as a standalone endpoint, infarct size by MRI at days 14–21, and ordinal shift in mRS. It is important to note that this revision occurred after the trial had concluded, raising serious concerns about changes to both the primary outcome and analysis methods after data unblinding. The sample size calculation was based on the original primary outcome as a continuous outcome and was not updated to reflect the revised binary outcome(s). The study is likely underpowered for many binary outcomes.

2. The IVT subgroup analysis was not pre-specified in the SAP. As a post hoc analysis, it is vulnerable to data-driven inference, including cherry-picking and inflated type I error rates.

3. The SAP explicitly states that missing data should be imputed with the worst possible score (e.g., mRS = 6, NIHSS = 42), but this is not described in the manuscript. In addition, baseline characteristics of participants lost to follow-up should be compared with those who completed the study to assess potential bias due to missing data.

4. The SAP indicates that an interim safety analysis was to be performed after the enrollment of 70 participants, but this is not mentioned in the manuscript. Reporting the conduct and results of the interim analysis is important for transparency and to confirm that continuation of the trial was not conditional on unreported findings.

5. The trial registration dated 2024-09-12 states that blood–brain barrier (BBB) permeability would be assessed via CT perfusion from Day 0 to Day 90. Although the SAP lists this as a secondary imaging outcome, no results or justification for its omission are provided in the manuscript.
6. The SAP specifies that transcriptomic and biomarker data were to be collected at multiple timepoints. However, the manuscript only refers to these briefly without providing details or results.
7. The manuscript states that data are available upon request but does not specify what data will be shared, how requests will be evaluated, or whether a data-sharing platform or repository will be used.
8. The manuscript does not describe how fidelity to the intervention (e.g., timing and adherence to semaglutide administration) was monitored or ensured.
9. Table 1 should report baseline characteristics rather than outcomes, and it would benefit from reorganization and removal of marginal p-values.
10. The CONSORT diagram should adhere more closely to established CONSORT reporting guidelines.

Version 2:

Reviewer comments:

Reviewer #3

(Remarks to the Author)

Authors are adequately answers to the questions/comments

Reviewer #4

(Remarks to the Author)

1.(Line 52) The primary endpoint should be clearly specified here, identifying which outcome serves as the main endpoint.
11. (Line 293) The endpoint configuration here is confusing—is safety or efficacy designated as the primary endpoint? This ambiguity must be clarified to ensure methodological rigor.

I acknowledge the author's detailed explanation regarding the primary endpoint, which clarifies their main intent. However, the description of the safety endpoint remains ambiguous: why is it termed 'primary safety endpoint'? What does 'primary' imply here? Is there prespecified statistical power for this endpoint? Literally, it appears equivalent to the mRS assessment. Moreover, this modification contradicts the protocol description. In the following protocol description, dual primary endpoints are still designated.

2.5. Endpoint measurement

Primary efficacy outcome is D90 mRS, mRS 0-2 is considered good outcome, while mRS 3-6 is considered poor outcome.

Primary safety outcome is a composite of death, MBE and ICH.

Furthermore, the author proposes a composite primary endpoint, which should be used with caution. The clinical significance of endpoints such as death and malignant edema may not be equivalent, and it is unclear whether they are assessed at a single follow-up time point.

Reviewer #6

(Remarks to the Author)

The authors have adequately addressed the critiques from the previous reviewers. I have no further concerns.

Version 3:

Reviewer comments:

Reviewer #4

(Remarks to the Author)

I believe the concerns have been fully addressed. I have no further comments at this time.

General Responses and Overview of Revision

We are grateful for the comments and constructive criticisms from the reviewers, which have helped us greatly when revising the manuscript. These include extensive revisions of the main text and references, inclusion of new analysis and crucial supplemental information, and better acknowledgement of study limitations.

Key major revisions include the following:

- Clarification of study outcomes and rationale: We have clarified the primary efficacy and safety outcomes throughout the manuscript to remove ambiguity. The Introduction has been substantially revised to strengthen the scientific rationale, incorporating key preclinical evidence in line with the STAIR X criteria, and elaborating on the proposed mechanisms of action for GLP-1RAs in acute ischemic stroke and upon reperfusion. These provide more clearly explained justifications for our Phase II trial and its design.
- Enhanced methodological transparency: The Methods section has been expanded to thoroughly outline the statistical models and clarify the analyses, including assumption testing, covariate adjustment, handling of missing data, and sensitivity analyses. We have also clarified the basis for the sample size calculation.
- Improved data presentation and reporting of results: For greater transparency, we now report on the pre-specified interim safety analysis, and have addressed all queries regarding the trial registration and statistical analysis plan. The CONSORT flow diagram and data tables have been revised for improved clarity and adherence to reporting guidelines.
- Strengthened discussion and acknowledgment of limitations: The Discussion has been carefully revised to better contextualize our findings. We have also more explicitly acknowledged the study's limitations, emphasizing the exploratory nature of interaction and subgroup analyses and positioning the findings as hypothesis-generating for a definitive Phase III trial.

We are confident that these and other revisions as detailed in the following parts of the rebuttal letter have substantially improved the manuscript, making it a stronger contribution to the field. Please see the following point-by-point responses to each comment.

Point-by-point Responses

Reviewer #3 (Remarks to the Author):

The work by Hao Wang and colleagues attempts addressing the safety and potential efficacy of GLP-1RA Semaglutide to improve neurological recovery in Large Vessel Occlusion stroke patients Treated by Reperfusion Therapy – in a Phase 2 Randomized Trial. This, in light of a possible Phase 3 clinical trial.

The work is timely and very relevant after a vast number of successful preclinical studies showing neuroprotection and improved functional recovery by GLP-1R agonists. However, some points need to be clarified to improve the manuscript.

General response

We are very thankful for the reviewer's encouraging comment that our work is timely and very relevant for the field, and giving us many helpful advice to improve the manuscript. Please see our detailed responses and revisions to address the reviewer's comments below.

1) The Introduction does not clearly motivate and explain the rationale of the study.

We sincerely appreciate the reviewer's valuable feedback. In response, we have revised the Introduction in accordance to points 1a-e (please see our detailed responses below). The Introduction now more clearly explains the study's motivation and rationale.

a. References 1-2 should be replaced with specific references related to GLP-1R expression in the brain (the original works). Moreover, it should be mentioned that the permeability of the brain to these drugs is uncertain (and provide references) and that CNS effects could be obtained indirectly, either to modulate normal physiology or in pathological conditions such as stroke.

We thank the reviewer for the suggestion. We have replaced references 1-2 with the original works related to GLP-1R expression in both rodent and human brains as advised. Moreover, we have highlighted that despite the variable permeability of different GLP1-RAs across the blood-brain barrier, preclinical studies found highly consistent neuroprotective effects observed in experimental stroke models given either before, at the time of, or even 24–72 hours after experimental stroke induction in various animal models (page 4, paragraph 1; please also see response to point 1b).

We have also elaborated the potential direct and indirect effects of GLP-1RAs on the central nervous system, and underlying mechanisms during acute stroke (page 4, paragraph 2): *“Potential neuroprotective mechanisms of GLP-1RAs in acute ischemic stroke can be direct or indirect. Direct effects include attenuation of detrimental microglia activation, promotion of M2 microglia polarization, reduction of neuronal apoptosis and promotion of neurogenesis, whereas indirect effects include amelioration of BBB breakdown, stabilization of blood glucose, modulation of systemic inflammation, and mimicking of ischemic preconditioning.”*

b. References 8-10 should be replaced/completed with the first original works showing the neuroprotective efficacy of GLP-1RA in animal models of stroke.

i. Li et al Proc Natl Acad Sci U S A. 2009 (in naïve rodents)

ii. Darsalia et al Cli Sci 2012 (in diabetic rodents)

iii. Moreover, the first and only available work showing neurological efficacy of GLP-1RA when given post-acute (from Day 3 after stroke) was provided by Augestad et al Br J Pharmacol 2012 and this study may be cited since strictly relevant for the ms.

We would like to thank the reviewer for the valuable suggestion. We have incorporated the recommended references into the revised manuscript as indicated (references 9, 10 & 8 in the revised manuscript, respectively), and revised the manuscript to better highlight the preclinical evidence for use of GLP-1RAs in acute ischemic stroke (page 4, paragraph 1): *“These neuroprotective effects generalized across GLP-1RAs with variable blood-brain barrier (BBB) permeability (attributable to differences in properties such as molecular size, albumin binding, and potentially receptor-mediated transport into the brain) and diverse experimental conditions, including various rodent species and strains, animal models without or with diabetes mellitus, stroke induction methods, treatment doses and regimens (initiating prior to, during, or even 1–3 days after stroke induction), indicating a robust class effect.”*

c. The reference 12 should be commented in more detail: I do not think that this study was designed to assess the neuroprotective efficacy/improve clinical outcome.

We thank the reviewer for the comment. The original reference 12 reported the TEXAIS trial (see *reference* below). The primary outcome of the trial was early neurological recovery, defined as an improvement of the National Institutes of Health Stroke Scale (NIHSS) of ≥ 8 (or NIHSS scores 0–1) at 7 days poststroke, in acute ischemic stroke patients who presented within a 9-hour window treated with exenatide vs. standard care. The safety outcomes included death, hyperglycemia, hypoglycemia, and other adverse events. The trial found no statistically significant difference in the primary outcome between the exenatide and standard care groups (61.2% vs. 56.7%, $p = 0.38$), although there was a significant reduction in hyperglycemic events (0.6/day vs. 0.9/day, $p = 0.05$).

We have revised and expanded this in the manuscript to provide a more detailed description of the TEXAIS trial and its goals as advised (page 10, paragraph 3): “*Thus far, only the phase 2 TEXAIS (Treatment with Exenatide in Acute Ischemic Stroke) randomized trial has been conducted, with the primary outcome defined as NIHSS improvement ≥ 8 (or NIHSS scores 0–1) at 7 days post-stroke. Focusing on this outcome measure, TEXAIS did not demonstrate benefit with exenatide in patients with acute ischemic stroke within 9 hours of onset, despite reporting a reduction in hyperglycemic events with exenatide.*”

Reference

Bladin CF, Wah Cheung N, Dewey HM, Churilov L, Middleton S, Thijs V, Ekinici E, Levi CR, Lindley R, Donnan GA, Parsons MW, Meretoja A, Tiainen M, Choi PMC, Cordato D, Brown H, Campbell BCV, Davis SM, Cloud G, Grimley R, Lee-Archer M, Ghia D, Sanders L, Markus R, Muller C, Salvaris P, Wu T, Fink J; TEXAIS Investigators. Management of Poststroke Hyperglycemia: Results of the TEXAIS Randomized Clinical Trial. *Stroke*. 2023 Dec;54(12):2962-2971.

d. The reference 13 should be checked since I am unsure if it supports the statement made.

We thank the reviewer for highlighting this oversight. Reference 13 has been removed from paragraph 2 of the Introduction as it was not relevant to the corresponding statement.

e. The text end of Intro only considers possible direct effect of Semaglutide in the brain after EVT. However, as commented above, the bbb may not allow the passage of GLP-1RA and thus alternative indirect ways may be added to the hypothesis of the work.

Thank you for the comment. Apart from acknowledging potential direct and indirect biological effects on the central nervous system (see response to point 1a above), we have now emphasized the possibility of alternative indirect mechanisms of action by semaglutide at the end of the **Introduction** (page 5, paragraph 1): “*In large vessel occlusion (LVO) stroke patients with salvageable penumbra, successful reperfusion by endovascular therapy (EVT) may enable the therapeutic potential of GLP-1RAs to be manifested by: 1) enabling better delivery of the drug and its secondary mediators to vulnerable brain tissues with ischemic injury (even though this is yet to be verified in animal models comparing drug distribution without and with reperfusion under otherwise identical experimental conditions), 2) counteraction of reperfusion injury and neuroinflammation once blood flow is restored, and 3) stabilization of the BBB. Apart from enhancing the rescue and protection of ischemic penumbra, these*

neurovascular benefits could also synergize with the systemic anti-inflammatory and glucose-stabilizing properties of GLP-1RAs to improve stroke outcomes.”

2) Results section

a. The Trial Population section in Results should be improved with a clear Study Design to complement Figure 1 or to add to Figure 1. It should be clearer the trial flow with also IVT treated patients.

We thank the reviewer for this valuable suggestion. As advised, we have included an overview of the study design in the first paragraph of the **Results** section to enhance clarity and readability. Additionally, we have revised **Figure 1** to more clearly illustrate the trial flow, specifying patients with or without intravenous thrombolysis:

b. It is well known that confounding factors for GLP-1RA treatment are glucose, blood pressure and BMI. The authors should expand on whether and how these factors have been handled in the study in both Results and Discussion.

Thank you for highlighting the potential confounding effects of blood glucose, blood pressure, and body mass index (BMI) in the context of GLP-1RA treatment (*reference 1*). We recognize that GLP-1RA can modulate these factors, which may in turn contribute to its overall therapeutic effect. Our primary analysis was to estimate the total effect of GLP-1RA, acknowledging that this may include indirect effects mediated through changes in glucose, blood pressure, and BMI – particularly the stabilization of blood glucose, as addressed in our responses to points 1a and 1e. Due to the limited number of primary efficacy outcome events (modified Rankin Scale [mRS] 0–2 at 90 days, $n = 78$), we pre-specified seven covariates for

adjustment in our primary analysis: premorbid modified Rankin Scale, National Institutes of Health Stroke Scale at presentation, modified Thrombolysis in Cerebral Infarction reperfusion grading, Alberta Stroke Program Early Computed Tomography Score, age, intravenous thrombolysis, and last-known-well-to-puncture time. These covariates were selected based on their strong associations with stroke outcomes (*reference 2 & 3*) and outlined in our statistical analysis plan. In order to balance statistical power, minimize collinearity and overfitting, and adhere to our pre-specified analysis plan, we did not include glucose, blood pressure, or BMI as covariates in the primary statistical models.

We appreciate that modulation of blood glucose, blood pressure, and body mass index may act as mediators of the potential treatment effect of GLP-1RA. However, as there was no statistically significant difference in the primary efficacy outcome between semaglutide and standard therapy in the overall population, further mediation analysis was not pursued. Additionally, neither diabetes mellitus nor blood glucose level on admission showed a statistically significant interaction with the treatment effect of GLP-1RA in our exploratory analyses (page 8, paragraph 2). Notably, among patients who did not receive intravenous thrombolysis, blood glucose level on day 3 was lower with semaglutide (6.4 ± 1.4 mmol/L vs. 7.8 ± 3.1 mmol/L), while the change between day-3 and baseline blood glucose with semaglutide was modest (-0.4 ± 1.2 mmol/L vs. 0.2 ± 2.7 mmol/L) (page 9, paragraph 2). We anticipate that our upcoming phase 3 trial with an expanded sample size will provide sufficient power for the suggested analyses.

Reference

1. Zheng Z, Zong Y, Ma Y, Tian Y, Pang Y, Zhang C, Gao J. Glucagon-like peptide-1 receptor: mechanisms and advances in therapy. *Signal Transduct Target Ther.* 2024 Sep 18;9(1):234.
2. Ramos LA, Kappelhof M, van Os HJA, Chalos V, Van Kranendonk K, Kruyt ND, Roos YBWEM, van der Lugt A, van Zwam WH, van der Schaaf IC, Zwinderman AH, Strijkers GJ, van Walderveen MAA, Wermer MJH, Olabarriaga SD, Majoie CBLM, Marquering HA. Predicting Poor Outcome Before Endovascular Treatment in Patients With Acute Ischemic Stroke. *Front Neurol.* 2020 Oct 15;11:580957.
3. Ospel JM, Ganesh A, Kappelhof M, McDonough R, Menon BK, Almekhlafi M, et al. Evaluating Outcome Prediction Models in Endovascular Stroke Treatment Using Baseline, Treatment, and Posttreatment Variables. *Stroke: Vascular and Interventional Neurology.* 2021;1(1):e000167.

3) Discussion

a. In relation to reference 14, see comment above when in the study by Augestad et al GLP-1RA was given from day 3 post-stroke

We thank the reviewer for the suggestion. In relation to the animal studies that demonstrated neurological benefits by GLP-1RA administered after stroke induction, we described the study by Augestad et al which GLP-1RA was given 3 days after stroke in **Discussion** (page 10, paragraph 2): “*Notably, the potential neuroprotective effects of GLP-1RA were evident even with delayed drug administration. In normoglycemic mice, a two-week course of daily liraglutide initiated 24 hours after stroke induction reduced infarct size, improved neurological recovery, and promoted angiogenesis. In another study involving diabetic mice, delayed administration of GLP-1RA 3 days after stroke induction for a duration*

of 8 weeks led to improved motor recovery and normalization of micro-vessel density and pericyte coverage”

b. Pls clarify better the following statement in Discussion page 9 “On the one hand, any potential direct interference of GLP-1RA activity by tenecteplase or alteplase, or vice versa, should be ruled out”.

Thank you for the comment. We have revised the statement to enhance the clarity (page 11, paragraph 2 to page 12, paragraph 1): *“Any potential pharmacokinetic interaction between GLP-1RA and tissue plasminogen activators should be excluded. On the other hand, GLP-1RA may inhibit the expression of plasminogen activator inhibitor-1 (PAI-1), a molecule that inhibits tissue plasminogen activators and the conversion of plasminogen to plasmin³⁷. Downregulation of PAI-1 with co-administration of IVT may thus indirectly potentiate plasmin generation and undesired effects of thrombolytic agents, such as BBB disruption, brain edema³⁸, and neurotoxicity. Nonetheless, these postulations require further confirmation.”*

c. Acknowledge the fact of the limited number of patients and whether, if known, the results will be sufficient to trigger a phase 3.

We appreciate the reviewer’s comment. We have highlighted the sample size limitation and the exploratory nature of the analysis involving the no-IVT stratum in **Discussion**, we believe the preliminary observation helps inform the design of a phase 3 trial to identify the patients that may be best treated by GLP-1RA in the context of LVO undergoing EVT (page 12, paragraph 1): *“Owing to the limited sample size inherited to the phase 2 trial design and the exploratory nature of this analysis, the possibility of type I error cannot be excluded. Thus, an adequately powered phase 3 study is needed to explore whether the potential neuroprotective effect of GLP-1RA is more pronounced in strokes with later presentations.”* We also further acknowledge that (page 12, paragraph 2): *“Our study has several limitations. First, the current phase 2 trial was not powered to draw definite conclusions regarding the efficacy of semaglutide in improving neurological outcomes in patients with LVO, but rather to estimate any potential treatment effects and evaluate the safety and tolerability of semaglutide treatment to inform the design for a large phase 3 trial.”*

d. In the results section the authors state: “The onset to puncture time was shorter in the IVT stratum compared to the no-IVT stratum (262 [171-397] minutes vs. 359 [248-521] minutes, p=0.002)” – an almost 2-hour difference. Timing of intervention is the most important predictor of neuroprotective efficacy. This should be discussed. Is it possible that in IVT strata, the maximum effect on efficacy outcome was already achieved by timely reperfusion and thus the modifying effect of Semaglutide was not detectable? – this should be discussed. The authors should also go deeper and analyze the effect of intervention time on efficacy outcome.

We sincerely appreciate the reviewer’s insightful comment. Indeed, earlier presentation may attenuate the potential effect of semaglutide, as neuroinflammation and other injury mechanisms in ischemic injury that could be counteracted with semaglutide may be more limited during the early time window, and efficacy outcomes could be primarily achieved through reperfusion with EVT.

In our preliminary data analysis based on restricted cubic spline, we found differing relationships between the risk ratio for achieving a good neurological outcome and last-known-well (LKW)-to-puncture time in the semaglutide vs. standard therapy arms (see *Figure* below). Semaglutide appeared to attenuate the time-dependency of EVT outcome ($p = 0.646$), whereas the outcome in the standard therapy arm was highly time-dependent ($p = 0.021$). This has been elaborated in the revised manuscript (page 9, paragraph 3): “*Modified Poisson regression with restricted cubic spline found differing relationships between LKW-to-puncture time and risk ratio of achieving the primary efficacy outcome in the semaglutide and standard therapy arms (Figure S1). In the standard therapy group, a longer onset-to-puncture time was associated with a lower likelihood of the primary efficacy outcome, whereas this association was diminished in the semaglutide group.*”

Figure S1

While this observation supports our findings comparing the IVT vs. no-IVT strata, we remain cautious due to the exploratory nature of the analysis. We therefore only include it as supplementary information in the manuscript, stating in **Methods** clearly that the analysis was not pre-specified (page 18, paragraph 1): “*Modified Poisson regression with restricted cubic spline was used to depict the relationship between LKW-to-puncture and risk ratios of achieving the primary efficacy outcome. All secondary outcomes and post-hoc analyses should be considered exploratory.*”

We fully acknowledge the exploratory nature of this analysis, and we are open to the reviewers’ and the editor’s further advice on whether this is the most optimal presentation.

In addition, we have revised the manuscript to provide a clearer account on this postulation (page 11, paragraph 2): “*As both neuroinflammation and reperfusion injury are time-dependent in ischemic stroke, the therapeutic benefits of GLP-1RAs in modulating or counteracting these processes may only become more apparent beyond the conventional 4.5-hour time window for IVT administration, during which the maximum treatment efficacy is primarily determined by reperfusion therapy.*”

Reviewer #4 (Remarks to the Author):

This study explores the innovative application of a Glucagon-Like Peptide-1 Receptor Agonist as an adjunctive neuroprotective therapy in patients undergoing

reperfusion treatment, a topic of significant clinical interest. However, the manuscript's scientific rigor is compromised by critical ambiguities in study design and endpoint definitions. A central issue lies in the unclear designation of the primary endpoint. The authors simultaneously emphasize both safety (e.g., symptomatic intracerebral hemorrhage) and efficacy (e.g., functional independence) as dual primary endpoints without hierarchical prioritization or statistical adjustments for multiplicity. This dual focus creates interpretative challenges: if efficacy is the primary objective, the absence of foundational Phase I data—such as pharmacokinetics (e.g., blood-brain barrier penetration), dose-escalation studies, or biomarker evidence of neuroprotection—undermines the rationale for directly advancing to a Phase II trial. Conversely, if safety is prioritized, the sample size justification based on efficacy outcomes (e.g., functional independence rates) becomes methodologically inconsistent, as safety endpoints typically require distinct statistical assumptions (e.g., non-inferiority margins for adverse events versus superiority for clinical outcomes). To strengthen the manuscript, the authors must clarify the study's phase and primary objective.

General response

We are deeply appreciative of the reviewer's very careful scrutiny of our study, and reminding us of the importance of clearly specifying the study's scientific and clinical basis for justifying a direct drug-repurposing Phase II trial, as well as clarifying the main objective to ensure methodological consistency. Please see our detailed responses and revisions to address each of the reviewer's comments below.

I have the following comments which the authors should address

1. (Line 52) The primary endpoint should be clearly specified here, identifying which outcome serves as the main endpoint.

We thank the reviewer for this valuable comment. We have clarified in the revised **Abstract** that our primary outcome is efficacy, defined as favorable neurological recovery (modified Rankin Scale 0–2 at 90 days). To avoid ambiguity, we have specified this as the main endpoint and have limited the reporting of *p*-value to the comparison of the primary efficacy outcome between semaglutide and standard treatment but not the safety outcome.

The revised **Abstract** now states that: *“The primary outcome was efficacy, defined as favorable neurological recovery (modified Rankin Scale 0–2 at 90 days). The primary safety outcome was a composite of death, malignant brain edema, and intracranial hemorrhage. Between August 2023 and July 2024, 140 patients were randomized to semaglutide (n=69) or standard therapy (n = 71). The primary efficacy outcome occurred in 39 (56.5%) in the semaglutide group and 39 (54.9%) in the standard therapy group (adjusted RR 1.05, 95% CI 0.95–1.15, p = 0.37). The primary safety outcome occurred in 16 (23.2%) in the semaglutide group and 17 (23.9%) in the standard therapy group (adjusted RR 0.99, 95% CI 0.89–1.11).”*

2. (Line 71) The authors claim that the drug aligns with STAIR criteria; however, they have not sufficiently demonstrated the relevant preclinical research progress, such as dose-response relationships, studies in animals of different sexes, different model or data from aged animal models.

We thank the reviewer for this critical point regarding the preclinical evidence and its alignment with the STAIR X criteria. We agree that the STAIR X guidelines provide an important framework for enhancing the rigor and translational potential of stroke research. Our decision to advance a GLP-1RA to a clinical trial was based on assessments of the totality and convergence of a large body of preclinical evidence, which we believe strongly satisfies the goals of STAIR X. This was thoroughly reviewed by and summarized in Maskery MP et al. (*reference 1*), and include the following:

- Dose-response relationships for GLP-1RAs in stroke are supported by key preclinical studies. For example, Darsalia V et al. (*reference 2*) demonstrated a dose-dependent reduction in infarct volume with increasing doses of exenatide (0.1, 1 or 5 µg/kg bw twice daily from 4 weeks prior to 2–4 weeks post-stroke induction in rat). Complementing this, Abdel-latif RG et al. (*reference 3*) examined dose-dependent effects for lixisenatide (at 1 and 10 nmol/kg bw given at 1 and 24 hours after reperfusion). While the two different lixisenatide doses were found to be similarly efficacious, Abdel-latif RG et al. (*reference 3*) verified that the neuroprotective effect of the low dose was abolished by a GLP-1R antagonist, confirming GLP-1R-mediated pathway engagement.
- The neuroprotective effects of GLP-1RAs are not confined to a single animal model. Their efficacy has been shown across multiple species and strains, including C57BL/6 mice, Sprague-Dawley rats, Wistar rats, and gerbils. Furthermore, efficacy was demonstrated in several stroke models, including both transient and permanent middle cerebral artery occlusion and common carotid artery occlusion, highlighting the robustness of the findings. These have been thoroughly reviewed and summarized by Maskery MP et al. (*reference 1*).
- While most preclinical studies used young adult animals, GLP-1RAs' efficacy in stroke has been shown in relevant comorbid animal models, such as the Goto-Kakizaki and the streptozotocin-induced diabetic rat models (see *reference 1* for details). Notably, Darsalia V et al. (*reference 4*) also found neuroprotective effects with exenatide treatment in aging/aged (14-month-old), obese, diabetic mice subjected to transient middle cerebral artery occlusion.
- A major strength of GLP-1RA based on the preclinical evidence is the validation of clinically relevant treatment paradigms. In animal studies, a wide range of dosing regimens have been tested, including prior to stroke onset (15 or 30 mins, or up to 7–14 days), following stroke onset (immediate, one to a few hours, or continued for half to 14 days), and combinations of above (see *reference 1* for details). Crucially, Chen Y et al. (*reference 5*) and Augestad et al. (*reference 6*) demonstrated neuroprotection with GLP-1RA treatment even when the first dose was delayed by 24 hours and 3 days post-stroke induction, respectively, mimicking a feasible clinical treatment window.
- We acknowledge that the vast majority of preclinical stroke studies were conducted in male animals, a recognized limitation across the field. However, we believe this preclinical evidence gap is substantially mitigated by clinical data. GLP-1RAs have been evaluated in large-scale trials involving thousands of diabetic and/or obese patients of both sexes, and often in older populations with other vascular comorbidities that increase stroke risk, where similar efficacies in male and female patients were found. Although this does not directly address the potential generalization when applied to acute stroke treatment, the extensive clinical data provides strong reassurance that the fundamental

biological response to GLP-1R agonism is consistent across sexes, making a sudden divergence in the acute ischemic stroke setting more unlikely.

In summary, while we agree that the preclinical evidence was not generated as a single, prospectively designed STAIR-X-compliant package, the convergent evidence is highly compelling. Data across many studies by different teams over the past two decades showed a class-wide, dose-dependent neuroprotective effect across multiple stroke models, in relevant aged/comorbid animals, and with clinically relevant treatment regimens. We thus concluded that GLP-1RAs represent one of the most promising classes of neuroprotective drug candidates for a clinical stroke trial. We have now revised the Introduction to better summarize the above-mentioned preclinical evidence and contextualise our rationale within the STAIR X framework. Please also see the responses to points 1a-e raised by Reviewer #3 for further details of the revision.

Reference

1. Maskery MP, Holscher C, Jones SP, Price CI, Strain WD, Watkins CL, Werring DJ, Emsley HC. Glucagon-like peptide-1 receptor agonists as neuroprotective agents for ischemic stroke: a systematic scoping review. *J Cereb Blood Flow Metab.* 2021 Jan;41(1):14-30.
2. Darsalia V, Mansouri S, Ortsäter H, Olverling A, Nozadze N, Kappe C, Iverfeldt K, Tracy LM, Grankvist N, Sjöholm Å, Patrone C. Glucagon-like peptide-1 receptor activation reduces ischaemic brain damage following stroke in Type 2 diabetic rats. *Clin Sci (Lond).* 2012 May 1;122(10):473-83.
3. Abdel-Latif RG, Heeba GH, Taye A, Khalifa MM. Lixisenatide ameliorates cerebral ischemia-reperfusion injury via GLP-1 receptor dependent/independent pathways. *Eur J Pharmacol.* 2018 Aug 15;833:145-154.
4. Darsalia V, Hua S, Larsson M, Mallard C, Nathanson D, Nyström T, Sjöholm Å, Johansson ME, Patrone C. Exendin-4 reduces ischemic brain injury in normal and aged type 2 diabetic mice and promotes microglial M2 polarization. *PLoS One.* 2014 Aug 7;9(8):e103114.
5. Chen Y, Zhang X, He J, Xie Y, Yang Y. Delayed Administration of the Glucagon-Like Peptide 1 Analog Liraglutide Promoting Angiogenesis after Focal Cerebral Ischemia in Mice. *J Stroke Cerebrovasc Dis.* 2018 May;27(5):1318-1325.
6. Augestad IL, Dekens D, Karampatsi D, Elabi O, Zabala A, Pintana H, Larsson M, Nyström T, Paul G, Darsalia V, Patrone C. Normalisation of glucose metabolism by exendin-4 in the chronic phase after stroke promotes functional recovery in male diabetic mice. *Br J Pharmacol.* 2021 Jun 16;179(4):677-694.

3. (Line 83/84) Does research evidence exist to support the claims that reperfusion improves the distribution of this drug and enhances its anti-inflammatory effects?

We thank the reviewer for raising this important question. Our postulation that administration of an experimental neuroprotective drug treatment with reperfusion may offer a better therapeutic window is based on several principles:

- First, restoring blood flow facilitates a systemically administered drug to influence the ischemic brain territory. No matter if an GLP-1RA acts directly on brain tissue or indirectly via systemic anti-inflammatory and metabolic modulatory effects, it is

plausible that reperfusion is essential to allow the circulating drug or other drug effect mediators to reach the site of ischemic injury.

- Second, a significant driver of ischemic tissue damage is reperfusion injury, characterized by acute inflammation and oxidative stress. The anti-inflammatory and anti-oxidative impacts of GLP-1RAs (e.g., as shown in various studies summarized in *reference 1* in the response to the previous point) are precisely suited to counteract this secondary injury. Administering semaglutide with reperfusion aligns its therapeutic action against this critical mechanism of brain tissue damage in stroke.
- Third, from a methodological standpoint, the potential neuroprotective effects of any experimental treatment can be masked if the ischemic deficit is too large (leaving little salvageable tissue) or too small (little room for improvement). In subjects with significant at-risk but viable penumbra as per our inclusion criteria (also see **Methods** and our response to comment 10), reperfusion creates the biological conditions under which any potential benefits of semaglutide are more likely to manifest.

Therefore, our core hypothesis is that reperfusion sets the stage for semaglutide to exert its effects, and for the benefits to be more detectable, thereby providing a viable clinical setting for the trial. We have now revised the manuscript to elaborate further on these points and explain our rationale in greater detail (page 4, paragraph 3 to page 5, paragraph 1). We also acknowledge that preclinical investigation directly comparing GLP-1RA efficacies and drug distribution with vs. without reperfusion under otherwise identical experimental stroke conditions is yet to be conducted (page 5, paragraph 1), which could be a reverse-translational follow-up study based on our clinical observations: *“To date, one phase 2 randomized trial demonstrated that exenatide, a GLP-1RA, did not improve early neurological recovery at 7 days post-stroke in a study population mainly consisting of mild strokes, despite a significant reduction of hyperglycemic events. The potential neuroprotective effects of GLP-1RAs remain unexplored in patients with a higher severity of stroke undergoing reperfusion therapy. In large vessel occlusion (LVO) stroke patients with salvageable penumbra, successful reperfusion by endovascular therapy (EVT) may enable the therapeutic potential of GLP-1RAs to be manifested by: 1) enabling better delivery of the drug and its secondary mediators to vulnerable brain tissues with ischemic injury (even though this is yet to be verified in animal models comparing drug distribution without and with reperfusion under otherwise identical experimental conditions), 2) counteraction of reperfusion injury and neuroinflammation once blood flow is restored, and 3) stabilization of the BBB. Apart from enhancing the rescue and protection of ischemic penumbra, these neurovascular benefits could also synergize with the systemic anti-inflammatory and glucose-stabilizing properties of GLP-1RAs to improve stroke outcomes.”*

4. (Line 96) In how many centers was the study conducted?

We thank the reviewer for the comment. The study was conducted across two thrombectomy centers in China. We have included an overview of the study design in the first paragraph of the **Results** section (page 5, paragraph 3): *“The GALLOP trial is a phase 2, investigator-initiated, prospective, randomized, open-label, blinded endpoint trial conducted at two thrombectomy centers in China, which also serve as the secondary referral sites from ten primary stroke centers.”*

5. (Line 110) All patients who received randomized allocation?

We thank the reviewer for the comment. The intention-to-treat population was defined as all patients who underwent randomization. We have explained more clearly in the revised text (page 7, paragraph 1): *“All analyses were based on the intention-to-treat population, defined as all patients who were randomized (n = 140). Patients who were lost to follow-up were assigned with the worst possible score for an outcome measure. Complete case analysis (n = 135) was performed for patients with no loss to follow-up as a sensitivity analysis.”*

6. (Line 111) It is challenging to draw confirmatory efficacy conclusions from this Phase II study.

We thank the reviewer for the comment, and we fully agree on the importance of clinical rigor when drawing conclusions. Indeed, it is challenging to draw definitive conclusions regarding efficacy from this Phase II study. We have addressed this limitation in greater detail in the revised manuscript in the following sections:

Abstract

“This phase 2 trial suggested semaglutide was safe in patients with LVO and was associated with an improved neurological outcome in patients not receiving IVT. These preliminary observations should be confirmed in a phase 3 randomized trial.”

Discussion (page 12, paragraphs 1–2):

“Owing to the limited sample size inherited to the phase 2 trial design and the exploratory nature of this analysis, the possibility of type I error cannot be excluded. Thus, an adequately powered phase 3 study is needed to explore whether the potential neuroprotective effect of GLP-1RA is more pronounced in strokes with later presentations.

Our study has several limitations. First, the current phase 2 trial was not powered to draw definite conclusions regarding the efficacy of semaglutide in improving neurological outcomes in patients with LVO, but rather to estimate any potential treatment effects and evaluate the safety and tolerability of semaglutide treatment to inform the design for a large phase 3 trial.”

7. (Line 119) This discrepancy is atypical. How did the authors define and adjudicate the occurrence of ICH (intracerebral hemorrhage)?

We thank the reviewer for the careful scrutiny of this clinical outcome. The occurrence of intracranial hemorrhage (ICH) was assessed on brain computed tomography performed three days after EVT or during clinical deterioration, which was elaborated in **Methods** (page 15, paragraph 2): *“All patients received a follow-up head CT and NIHSS reassessment three days after the procedure or during clinical deterioration to detect any intracranial hemorrhage (ICH) or malignant brain edema (MBE).”*

Furthermore, we clarified the definition of ICH with Heidelberg Bleeding Classification of 2 or above (see *Reference*) (page 16, paragraph 3): *“The primary safety outcome was a composite of death, MBE (parenchymal hypodensity of at least 50% of the MCA territory with signs of local brain swelling such as sulcal effacement and compression of the lateral ventricle, and midline shift of ≥ 5 mm at the septum pellucidum or pineal gland with obliteration of the*

basal cisterns), and any ICH with Heidelberg Bleeding Classification of 2 or above (intracerebral hemorrhage occupying more than 30% of the infarcted tissue with mass effect, and/or intracerebral hemorrhage outside the infarcted brain tissue, or any intracerebral-extracerebral hemorrhage, such as hematoma in parenchymal tissue remote from the infarcted area, intraventricular hemorrhage, subarachnoid hemorrhage, or subdural hemorrhage).”

In the revised manuscript, this definition is also stated in the **Results** section when describing the safety outcome (page 7, paragraph 2): *“The primary safety outcome, which is a composite of death, malignant brain edema (parenchymal hypodensity of at least 50% of the MCA territory with signs of local brain swelling such as sulcal effacement and compression of the lateral ventricle, and midline shift of ≥ 5 mm at the septum pellucidum or pineal gland with obliteration of the basal cisterns) and intracranial hemorrhage (ICH, indicated by intracranial bleeding of Heidelberg Bleeding Classification of 2 or above) occurred in 16 (23.2%) in the semaglutide group and 17 (23.9%) in the standard therapy group (adjusted RR 0.99, 95% CI 0.89–1.11).”*

The adjudication of post-EVT ICH was conducted centrally in our image processing laboratory. Radiological outcomes, including ICH, were determined by two experienced raters, each with over ten years of expertise. Any disagreements were resolved by a third independent rater. Adjudicators were blinded to treatment allocation and clinical outcomes to minimize bias. This is also clearly stated in the manuscript (page 16, paragraph 2): *“...hemorrhagic complications were determined by two raters (BI, JA) with more than 10 years of experience. Disparities were resolved by a third rater (SM). Raters and computer algorithms were blinded from treatment allocation and clinical outcomes.”*

Please see the detailed response to point 8 for the plausible explanations on ICH rate discrepancy.

Reference

von Kummer R, Broderick JP, Campbell BC, Demchuk A, Goyal M, Hill MD, Treurniet KM, Majoie CB, Marquering HA, Mazya MV, San Román L, Saver JL, Strbian D, Whiteley W, Hacke W. The Heidelberg Bleeding Classification: Classification of Bleeding Events After Ischemic Stroke and Reperfusion Therapy. *Stroke*. 2015 Oct;46(10):2981-6.

8. (Line 156) A symptomatic intracerebral hemorrhage (SICH) rate of 9.9% was observed in the control group, which is unusually high. What factors might explain this elevated incidence?

Thank you for highlighting the concern regarding the observed rate of symptomatic intracranial hemorrhage (ICH) in our control group, and the difference between the two treatment groups. Our central image processing laboratory thoroughly reviewed each digital subtraction angiography performed immediately after EVT and confirmed that none of the study participants exhibited contrast extravasation. Therefore, it is unlikely that the elevated rate of symptomatic ICH in the control group was attributable to periprocedural vessel perforation. This information has been updated in the manuscript (page 6, paragraph 2): *“No patients had contrast extravasation on selective internal carotid artery angiography performed immediately after EVT.”*

A heightened risk of symptomatic ICH among Chinese patients could be inherent to the greater prevalence of intracranial atherosclerosis (*reference 1*), which can increase procedure

complexity as reflected by a relatively higher rate of acute intracranial stenting – a potential risk for symptomatic ICH (*reference 2*). Nevertheless, we wish to point out that the occurrence rates for ICH in our study were in line with those reported in the literature. A large nationwide observational registry study ($n = 3077$; *reference 3*) reported an ICH frequency of 24.2%, with Heidelberg classification 2 or above occurring in approximately 17% of cases. These figures are similar to those observed in our control group. In the same study, the frequency of symptomatic ICH was 4.5%. However, several studies have reported a 9–16% risk of symptomatic ICH following EVT, including one within a randomized trial setting (*references 2 & 4*).

Regarding the observed difference in ICH rates between the semaglutide and control arms, preclinical evidence suggests that GLP1-RA can stabilize the blood-brain barrier and ameliorate infarcts with hemorrhagic transformation (*reference 5*). To date, only one clinical study has demonstrated a reduced incidence of intracerebral hemorrhage in diabetic patients treated with GLP1-RA compared to those receiving other hypoglycemic agents (*reference 6*). This finding indirectly supports the notion that GLP1-RA may reduce the risk of brain hemorrhages, potentially through protection of the blood-brain barrier or indirectly via stabilization of systemic glucose, a well-established risk factor for post-EVT intracranial hemorrhage (*reference 7*). Nevertheless, as this is a secondary outcome with a relatively low event rate, the possibility of a type I error cannot be excluded. In the revised manuscript, we have refrained from drawing strong conclusions and omitted the reporting of the nominal p -values for these analyses. Furthermore, we emphasize that, due to the Phase II study design, it is challenging to draw definitive conclusions (please see response to point 6).

Reference

1. Leng X, Hurford R, Feng X, Chan KL, Wolters FJ, Li L, Soo YO, Wong KSL, Mok VC, Leung TW, Rothwell PM. Intracranial arterial stenosis in Caucasian versus Chinese patients with TIA and minor stroke: two contemporaneous cohorts and a systematic review. *J Neurol Neurosurg Psychiatry*. 2021 Mar 30;92(6):590–7.
2. Hao Y, Yang D, Wang H, Zi W, Zhang M, Geng Y, Zhou Z, Wang W, Xu H, Tian X, Lv P, Liu Y, Xiong Y, Liu X, Xu G; ACTUAL Investigators (Endovascular Treatment for Acute Anterior Circulation Ischemic Stroke Registry). Predictors for Symptomatic Intracranial Hemorrhage After Endovascular Treatment of Acute Ischemic Stroke. *Stroke*. 2017 May;48(5):1203-1209.
3. Hall E, Ullberg T, Andsberg G, Wasselius J. Incidence of intracranial hemorrhagic complications after anterior circulation endovascular thrombectomy in relation to occlusion site: a nationwide observational register study. *J Neurointerv Surg*. 2024 Oct 14;16(11):1088-1093.
4. van der Steen W, van der Ende NAM, Luijten SPR, Rinkel LA, van Kranendonk KR, van Voorst H, Roosendaal SD, Beenen LFM, Coutinho JM, Emmer BJ, van Oostenbrugge RJ, Majoie CBLM, Lingsma HF, van der Lugt A, Dippel DWJ, Roozenbeek B; MR CLEAN-NO IV, MR CLEAN-MED, and CONTRAST investigators. Type of intracranial hemorrhage after endovascular stroke treatment: association with functional outcome. *J Neurointerv Surg*. 2023 Oct;15(10):971-976.

5. Chen F, Wang W, Ding H, Yang Q, Dong Q, Cui M. The glucagon-like peptide-1 receptor agonist exendin-4 ameliorates warfarin-associated hemorrhagic transformation after cerebral ischemia. *J Neuroinflammation*. 2016 Aug 26;13(1):204.
6. Pasi M, Bretonnière A, Lochon L, Dosda A, Bisson A, Boulouis G, Ducluzeau PH, Fauchier L. Glucagon-Like Peptide-1 Receptor Agonists and Risk of Nontraumatic Intracerebral Hemorrhage in Patients With Type 2 Diabetes. *Stroke*. 2025 Jun 5.
7. Tanaka K, Yoshimoto T, Koge J, Yamagami H, Imamura H, Sakai N, Uchida K, Beppu M, Matsumaru Y, Matsumoto Y, Kimura K, Ishikura R, Inoue M, Sakakibara F, Morimoto T, Yoshimura S, Toyoda K; RESCUE-Japan LIMIT Investigators. Detrimental Effect of Acute Hyperglycemia on the Outcomes of Large Ischemic Region Stroke. *J Am Heart Assoc*. 2024 Dec 3;13(23):e034556.

9. Heterogeneity across participating centers should be acknowledged as a limitation of this study

Thank you for the comment. We have acknowledged the heterogeneity across the participating centers as a limitation of our study (page 12, paragraph 2): “*Second, as randomization only took place in two thrombectomy centers, future studies should consider evaluating GLP-1RA administration at primary stroke centers prior to secondary transfer for EVT. Expanding the number of participating centers and accounting for heterogeneity across sites are also necessary to provide a more accurate estimate of the treatment effect and enhance the generalizability of the findings.*”

10. (Line 237) What were the specific inclusion criteria for hemodynamic parameters (e.g., cerebral blood flow [CBF], time-to-peak [TTP], mean transit time [MTT]) derived from perfusion imaging in this study?

We included patients whose last known well time was within 12 hours at presentation. Patients presenting within 6 hours were eligible if they had an occlusion of the terminal internal carotid artery or middle cerebral artery and an ASPECTS score of ≥ 6 . For those presenting between 6 and 12 hours, eligibility required an ASPECTS score of ≥ 6 , and evidence of a clinical-core or penumbra-core mismatch, as defined by the DAWN or DEFUSE3 criteria, respectively.

In the revised manuscript, we have included an overview of the study design that highlighted the radiological inclusion criteria, and stated the DAWN and DEFUSE3 criteria in the **Methods** and **Results** sections:

Results (page 6, paragraph 1):

*“Main eligibility criteria include: 1) LVO at the M1 segment of middle cerebral artery (MCA) or terminal internal carotid artery, 2) last-known-well (LKW) within 12 hours at presentation, 3) Alberta Stroke Program Early Computed Tomography Score (ASPECTS) ≥ 6 , and 4) for patients with a LKW between 6 and 12 hours at presentation, computed tomography (CT) perfusion demonstrating a significant clinical-radiological or core-penumbra mismatch according to the DAWN or DEFUSE3 criteria, respectively (see **Methods** for details)”*

Methods (page 13, paragraph 3 to page 14, paragraph 1)

“The DAWN criteria required a mismatch between clinical deficit and infarct core volume as determined by CT perfusion: patients aged ≥ 80 years needed an NIHSS score ≥ 10 and infarct core (volume of relative cerebral blood flow [rCBF] $< 30\%$) < 21 mL; those < 80 years needed NIHSS ≥ 10 and core < 31 mL, or NIHSS ≥ 20 and core 31 – < 51 mL. The DEFUSE3 criteria required imaging evidence of a target mismatch profile: infarct core (volume of rCBF $< 30\%$) < 70 mL, penumbral tissue (volume of $T_{max} > 6s$ - volume of rCBF $< 30\%$) ≥ 15 mL, and a mismatch ratio (volume of $T_{max} > 6s$: volume of rCBF $< 30\%$) of ≥ 1.8 .”

11. (Line 293) The endpoint configuration here is confusing—is safety or efficacy designated as the primary endpoint? This ambiguity must be clarified to ensure methodological rigor.

We appreciate the reviewer’s valuable feedback regarding the definitions of our primary efficacy and safety outcomes, and we apologize for any ambiguity in our initial manuscript. To clarify, our study’s primary endpoint is the efficacy outcome, defined as the proportion of patients achieving a modified Rankin Scale (mRS) score of 0–2 at 90 days. The primary safety outcome, which we have highlighted to underscore its clinical importance, is a composite of death, malignant brain edema, and intracranial hemorrhage (ICH). We wish to emphasize that the principal focus of our analysis is on the efficacy outcome, and the primary safety outcome is reported descriptively to inform on any potential risks with GLP-1RA treatment. Accordingly, throughout the revised manuscript we have refrained from reporting a *p*-value for the primary safety outcome, as it was not intended as a co-primary endpoint. We hope this clarification addresses the reviewer’s concerns, and we are grateful for the opportunity to improve the clarity of our manuscript.

12. (Line 307) The sample size justification is similarly unclear and lacks empirical foundation. If, as the authors state, no prior human studies have investigated this drug in clinical settings, it would be methodologically imperative to first conduct a first-in-human (FIH) Phase I trial to obtain critical safety data (e.g., dose-escalation findings), pharmacokinetic/pharmacodynamic (PK/PD) parameters, and preliminary evidence of biological activity. These data are essential for rationally estimating the sample size required for subsequent efficacy-focused studies (e.g., Phase II). The authors must provide further clarification on this process.

We thank the reviewer for raising this crucial point about ensuring patient safety and obtaining PK/PD data along with preliminary evidence of desired biological activity in stroke patients. We take this as an opportunity to further elaborate on, and clarify our justifications to launch the trial as a Phase II proof-of-concept (PoC) study, and why our approach aligns with established methodological pathways for drug repurposing.

We fully agree that for a new chemical entity, a Phase I trial is an absolute prerequisite to establish its safety and PK/PD profile. For semaglutide, its safety, tolerability, and PK/PD are exceptionally well-characterized from extensive clinical trials and real-world use in millions of patients. Crucially, such data came from diverse populations that include those that share the same characteristics as our target stroke population: older adults with type 2 diabetes, hypertension, and other cardiovascular risk factors or comorbidities, without contraindications

for GLP-1RAs. Therefore, the baseline safety and PK/PD in our target patient group was already understood in depth.

Regarding whether semaglutide is equally safe in acute stroke setting, part of our trial is indeed designed to address this very important question. Notably, GLP-1RAs have an exceptionally wide therapeutic window. Reported cases of overdosing liraglutide or semaglutide by tens of times above FDA-approved doses only presented with relatively mild, manageable gastrointestinal side effects (Nakanishi R et al., [reference 1] & Branch MR, et al. [reference 2]). Furthermore, the 0.5 mg dose we selected for semaglutide is conservative, sitting at the low end of the clinically established range (typically 0.25 to 2.4 mg, with 7.2 mg tested in the recently completed STEP UP Phase III trial). There is no compelling scientific rationale to suggest that its safety would be drastically altered by the pathophysiology of stroke. In line with this, the TEXAIS trial (Bladin CF et al., [reference 3]) provided direct clinical evidence for the safety of GLP-1RA in acute ischemic stroke patients. In that study, which evaluated the use of exenatide 5µg twice daily for 5 days in patients with acute ischemic stroke who presented within a 9-hour window, exenatide treatment did not result in an increase in severe adverse events either in the overall cohort or in the subgroup of patients ($n = 197$) who received intravenous thrombolysis and/or endovascular therapy. Importantly, our trial protocol includes rigorous safety monitoring, with mandatory reporting of adverse events (e.g., death, ICH, infection) to statutory bodies, including the Department of Health of Hong Kong, and respective clinical research ethics committees of the New Territories East Cluster of Hong Kong, and Linyi People's Hospital. The study protocol also incorporated intensive monitoring of neurological and systemic status before, during, and after semaglutide administration. This allowed us to gather safety data in the target patient group, while simultaneously evaluating potential efficacy.

Building on strong scientific grounds for safety and potential efficacy, we therefore positioned our study as an exploratory Phase II PoC trial, with the primary goal to determine if there is a signal of efficacy sufficient to justify a larger, definitive Phase III trial. We have revised the manuscript to explain more clearly our rationale for the study (in the **Introduction**, also see responses to major comments 2 & 3), and emphasize the trial's Phase II PoC nature by clearly stating that our main primary outcome was efficacy in **Abstract** and **Results** (e.g., see page 7, paragraph 2).

We performed sample size estimation for this Phase II PoC trial assuming a hypothesized, clinically meaningful effect size of 0.15–0.25 in the primary efficacy outcome. The sample size estimation was based on simulation studies (reference 4), which demonstrated that for trials with binary outcomes, the best balance between the gain in precision of event rate estimates and minimizing the size of the external pilot trial was 60 per group (120 in total). This finding is consistent across a range of plausible effect sizes (0.1 to 0.5), and is based on the use of robust statistical methods (reference 4). After considering a rate of 2.5% suboptimal scan quality, 5% loss to follow-up, and 10% of suboptimal recanalization according to the track records of participating centers, 140 participants were required. These have been described in detail in the **Study Protocol**.

Reference

1. Bin Nafisah S, Almatrafi D, Al-Mulhim K. Liraglutide overdose: A case report and an updated review. *Turk J Emerg Med.* 2020 Jan 28;20(1):46-49.

2. Branch MR, Amador IE, Tardif I, Patel KK, Lewis DA. Clinical Manifestations of Semaglutide Overdose: A Case Study. *J Psychiatr Pract*. 2024 Nov 1;30(6):444-446.
3. Bladin CF, Cheung NW, Dewey HM, Churilov L, Middleton S, Thijs V, Ekinçi E, Levi CR, Lindley R, Donnan GA, Parsons MW, Meretoja A, Tiainen M, Choi PM, Cordato D, Brown H, Campbell BC, Davis SM, Cloud G, Grimley R, Lee-Archer M, Ghia D, Sanders L, Markus R, Muller C, Salvaris P, Wu T, Fink J; TEXAIS Investigators. Management of Poststroke Hyperglycemia: Results of the TEXAIS Randomized Clinical Trial. *Stroke*. 2023 Nov 28;54(12):2962-2971.
4. Teare D, Dimairo M, Hayman A, Shephard N, Whitehead A, Walters S. Sample Size Requirements for Pilot Randomised Controlled Trials with Binary Outcomes: A Simulation Study. *Trials*. 2013 Nov 29;14(Suppl 1):O21.

13. Line 315 The statistical model employed for the primary endpoint must be thoroughly elaborated, including its rationale, assumptions, and adjustments. This is critical to ensure the validity and reproducibility of the findings. Key details—such as covariate adjustment, handling of missing data, and validation of model assumptions (e.g., normality, proportional hazards)—are omitted, undermining the credibility of the results.

The comparison of the primary efficacy endpoint (mRS 0–2 at 90 days) was performed using a modified Poisson regression model, i.e., Poisson regression with a robust error variance estimator to correct standard errors for binary data. The rationale of using modified Poisson regression was that, compared to logistic regression, modified Poisson regression could provide a more unbiased estimation of risk ratios in the setting of common events, in which odds ratio computed by logistic regression may overstate treatment effect size compared to risk ratios. Please refer to the point-by-point responses to the specific recommendations below for details.

a. Specific Recommendations:

Explicitly state whether the analysis used a generalized linear model (GLM), Cox proportional hazards model, mixed-effects model, or another approach. Provide the mathematical formulation if possible.

Thank you for your comment. We have explicitly stated the use of modified Poisson regression for the primary efficacy outcome in **Methods** (page 17, paragraph 4): “*We used the modified Poisson regression models to evaluate the risk ratios (RR) of the primary and binary secondary outcomes of semaglutide vs standard therapy.*”

Regarding the mathematical formulation, we fully recognize the importance of transparency and reproducibility. However, due to the length and complexity of the complete mathematical specification, we have instead provided a clear reference to the original work where the detailed formulation can be found in reference 50 of the revised manuscript (*reference*).

Reference

Zou, G. (2004). A modified poisson regression approach to prospective studies with binary data. *American Journal of Epidemiology* 159, 702-706.

b. Clarify if and how baseline covariates (e.g., age, baseline severity) were included. Were interactions tested?

Thank you for the important inquiry. In our modified Poisson regression analysis comparing the primary efficacy outcome between semaglutide and standard therapy, we included a pre-specified set of covariates: age, premorbid modified Rankin Scale (mRS), intravenous thrombolysis (IVT), National Institutes of Health Stroke Scale (NIHSS) at presentation, last-known-well to puncture time, modified Thrombolysis in Cerebral Infarction (mTICI) reperfusion grade, and baseline Alberta Stroke Program Early Computed Tomography Score (ASPECTS). These covariates were selected a priori due to their well-established clinical relevance to neurological recovery following EVT (*references 1 & 2*).

Interactions were examined for IVT, mTICI, diabetes mellitus, baseline blood glucose and baseline blood glucose level in exploratory analyses, based on the following considerations:

- IV thrombolysis and peptide drug interactions: Intravenous thrombolytic agents (such as alteplase and tenecteplase), as well as plasmin (enhanced formation from plasminogen by IVT) are serine proteases that cleave peptides and may inactivate some peptide-based drugs. Such drug-drug interaction has been demonstrated in vivo with nerinetide (*reference 3*), a peptide previously investigated for neuroprotection in the setting of acute large vessel occlusion. Although there is no experimental evidence showing cleavage of GLP-1 or its analogues by these serine proteases, for scientific rigor we could not rule out the possibility.
- Reperfusion (mTICI grading): Although both transient and permanent middle cerebral artery occlusion animal models indicate that GLP-1RA can improve stroke outcomes (*reference 4*), good reperfusion (as assessed by mTICI) may further enhance the therapeutic efficacy of GLP-1RA by improving drug or secondary mediator delivery to vulnerable brain tissues, once antegrade blood flow has been re-established.
- Diabetes mellitus and blood glucose: Patients with diabetes mellitus or elevated baseline blood glucose may benefit more from the systemic glucose-stabilizing effects of GLP-1RA treatment.

The detailed findings are presented in **Results** (page 8, paragraph 2): “*We looked for treatment effect modification by covariates pre-specified in the statistical model. We found that IVT significantly interacted with the treatment effect by semaglutide ($p_{interaction} = 0.02$). mTICI, and post-hoc analyses of diabetes mellitus and baseline blood glucose level did not have treatment effect modification with semaglutide.*”

The exploratory nature of these analyses was emphasized in the **Discussion** and **Methods** sections. Please see the response to point 13g for details.

Reference

1. Ramos LA, Kappelhof M, van Os HJA, Chalos V, Van Kranendonk K, Kruyt ND, Roos YBWEM, van der Lugt A, van Zwam WH, van der Schaaf IC, Zwinderman AH, Strijkers GJ, van Walderveen MAA, Wermer MJH, Olabarriaga SD, Majoie CBLM, Marquering HA. Predicting Poor Outcome Before Endovascular Treatment in Patients With Acute Ischemic Stroke. *Front Neurol.* 2020 Oct 15;11:580957.

2. Ospel JM, Ganesh A, Kappelhof M, McDonough R, Menon BK, Almekhlafi M, et al. Evaluating Outcome Prediction Models in Endovascular Stroke Treatment Using Baseline, Treatment, and Posttreatment Variables. *Stroke: Vascular and Interventional Neurology*. 2021;1(1):e000167.
3. Hill MD, Goyal M, Menon BK, Nogueira RG, McTaggart RA, Demchuk AM, Poppe AY, Buck BH, Field TS, Dowlatshahi D, van Adel BA, Swartz RH, Shah RA, Sauvageau E, Zerna C, Ospel JM, Joshi M, Almekhlafi MA, Ryckborst KJ, Lowerison MW, Heard K, Garman D, Haussen D, Cutting SM, Coutts SB, Roy D, Rempel JL, Rohr AC, Iancu D, Sahlas DJ, Yu AYZ, Devlin TG, Hanel RA, Puetz V, Silver FL, Campbell BCV, Chapot R, Teitelbaum J, Mandzia JL, Kleinig TJ, Turkel-Parrella D, Heck D, Kelly ME, Bharatha A, Bang OY, Jadhav A, Gupta R, Frei DF, Tarpley JW, McDougall CG, Holmin S, Rha JH, Puri AS, Camden MC, Thomalla G, Choe H, Phillips SJ, Schindler JL, Thornton J, Nagel S, Heo JH, Sohn SI, Psychogios MN, Budzik RF, Starkman S, Martin CO, Burns PA, Murphy S, Lopez GA, English J, Tymianski M; ESCAPE-NA1 Investigators. Efficacy and safety of nerinetide for the treatment of acute ischaemic stroke (ESCAPE-NA1): a multicentre, double-blind, randomised controlled trial. *Lancet*. 2020 Mar 14;395(10227):878-887.
4. Maskery MP, Holscher C, Jones SP, Price CI, Strain WD, Watkins CL, Werring DJ, Emsley HC. Glucagon-like peptide-1 receptor agonists as neuroprotective agents for ischemic stroke: a systematic scoping review. *J Cereb Blood Flow Metab*. 2021 Jan;41(1):14-30.

c. How were violations of model assumptions (e.g., non-linearity in logistic regression) addressed?

We addressed potential violations of model assumptions by conducting appropriate diagnostic tests for each regression analysis. For the modified Poisson regression model, we assessed the log-linear mean assumption and overall model fit using the normalized residual sum of squares (NRSS) test (*reference 1*), which yielded a *p*-value of 0.794, indicating no evidence of lack of fit. For ordinal logistic regression, we evaluated the proportional odds assumption using the Brant test (*reference 2*), obtaining a *p*-value of 0.66. These results indicate that there were no obvious violations of model assumptions in our analyses. We have specified these tests in the revised **Results** and **Methods**:

Results (page 7, paragraph 1):

“There were no violations of the statistical assumptions of the modified Poisson regression models and proportional odds assumption for the ordinal logistic regression.”

Methods (page 18, paragraph 1)

“The goodness-of-fit of the modified Poisson regression models was assessed using the normalized sum of square test. Proportional odds ordinal logistic regression was used to compare the ordinal shift of mRS between the two groups. The proportional odds assumption was tested with the Brant test.”

Reference

1. Hagiwara Y, Matsuyama Y. Goodness-of-fit tests for modified Poisson regression possibly producing fitted values exceeding one in binary outcome analysis. *Stat Methods Med Res*. 2024 Jul;33(7):1185-1196.

2. Brant R. Assessing proportionality in the proportional odds model for ordinal logistic regression. *Biometrics*. 1990 Dec;46(4):1171-8.

d. Describe sensitivity analyses (e.g., alternative models, different imputation methods; per-protocol analysis) to confirm the robustness of the primary endpoint results.

We thank the reviewer for the inquiry and we fully agree on the importance of ensuring the robustness of our findings. We have included sensitivity analysis based on ensuring consistent conclusions with alternative statistical models, a conservative imputation method, and per-protocol analysis:

- We compared the proportion of participants who achieved good neurological recovery (defined as mRS 0–2) between the semaglutide and standard therapy groups using *modified Poisson regression*. In addition, we applied *ordinal logistic regression* to evaluate the ordinal shift in mRS scores between the two groups. These analyses are detailed in the **Results** section (under the subheadings of *Primary Outcomes* and *Secondary Outcomes*), and give consistent results as illustrated in **Figures 2 & 3**.
- For imputation of missing outcome data, we adopted a conservative approach by assigning the worst possible score to affected participants: three in the semaglutide group and two in the control group were assigned an mRS of 6 (indicating death) and an NIHSS of 42 (the maximum score). No other missing data relevant to the study required further imputation. Details of this approach are provided in **Results** (page 6 paragraph 2): “*The mRS at 90 days was missing for 3 (4.3%) patients in the semaglutide group and 2 (2.8%) patients in the standard therapy group due to loss to follow-up. There was no other missing data.*”; and **Methods** (page 18, paragraph 1): “*Missing outcomes were imputed as the worst possible score for an outcome measure, i.e. an mRS of 6 and an NIHSS of 42.*”
- Finally, we conducted complete case analyses (see **Tables S8–S10**), excluding participants who were lost to follow-up. The per-protocol sample defined in the study protocol is identical to the complete case sample as no participants had clinically meaningful deviation from the protocol, apart from the five individuals who were lost to follow-up. Details of the minor protocol deviation are provided in **Table S3**.

e. Are all analyses based on the ITT population (intention-to-treat), as defined in the protocol?

Thank you for the question. All analyses, including the modified Poisson regression and ordinal logistic regression models, were conducted based on the intention-to-treat (ITT) population as defined in the study protocol, except for the complete case analyses as a sensitivity check. This has been clarified in the revised manuscript (page 7, paragraph 1): “*All analyses were based on the intention-to-treat population, defined as all patients who were randomized (n = 140). Patients who were lost to follow-up were assigned with the worst possible score for an outcome measure. Complete case analysis (n = 135) was also performed for patients with no loss to follow-up as a sensitivity analysis.*”

f. Was a safety set analysis omitted, and if so, justify its exclusion despite being standard in safety reporting.

Thank you for the comment regarding the inclusion of a safety set analysis. In this study, all patients randomized to the semaglutide arm received at least one dose of semaglutide. Consequently, the primary efficacy analysis set and the safety analysis set were identical. We have clarified this point in the revised manuscript (page 6, paragraph 2): *“All patients randomized to the semaglutide group received semaglutide injection before puncture, while 3 (4.3%) patients in the semaglutide group did not complete day-7 semaglutide injection due to mortality.”*

g. Are Multiplicity adjustments used in this analysis? Methods to control type I error inflation from multiple hypothesis testing.

We thank the reviewer for raising the important issue of multiplicity adjustment. Given the Phase II study design and the limited sample size, we have reported the p -value only for the statistical comparison of the primary efficacy outcome in the revised manuscript. This approach underscores the primary efficacy outcome as the main focus of our analysis, while all other analyses including the interaction between intravenous thrombolysis and treatment effect of semaglutide are presented as exploratory.

We believe this approach helps to mitigate the risk of type I error inflation and overinterpretation of the secondary/exploratory analyses. Additionally, we have emphasized the potential for type I error in exploratory analyses and highlighted that definite conclusions regarding the efficacy of semaglutide cannot be drawn due to the relatively small sample size in the revised manuscript (page 12, paragraphs 1–2): *“Owing to the limited sample size inherited to the phase 2 trial design and the exploratory nature of this analysis, the possibility of type I error cannot be excluded. Thus, an adequately powered phase 3 study is needed to explore whether the potential neuroprotective effect of GLP-1RA is more pronounced in strokes with later presentations. Our study has several limitations. First, the current phase 2 trial was not powered to draw definite conclusions regarding the efficacy of semaglutide in improving neurological outcomes in patients with LVO, but rather to estimate any potential treatment effects and evaluate the safety and tolerability of semaglutide treatment to inform the design for an adequately powered phase 3 trial.”*

Reviewer #5 (Remarks to the Author):

Reviewer #6 (Remarks to the Author):

This phase 2 randomized trial evaluated the safety and efficacy of semaglutide, a GLP-1 receptor agonist, in 140 Chinese patients with acute anterior circulation large vessel occlusion stroke treated with endovascular therapy (EVT). Semaglutide did not improve the primary efficacy endpoints, functional independence at 90 days (mRS 0–2)

and a composite safety outcome, compared to standard care, but it was significantly associated with lower rates of symptomatic ICH and better neurological recovery. An exploratory subgroup analysis suggested benefit in patients not receiving IV thrombolysis, though this was not pre-specified and is subject to inflated type I error. While the trial was well conducted and clearly reported, the clinical significance is limited given the null primary outcome, lack of adjustment for multiple comparisons, and post hoc changes. The findings can be viewed as hypothesis-generating and require validation in phase 3 trials before influencing practice. The manuscript is largely consistent with the statistical analysis plan (SAP) in its study design, eligibility criteria, intervention protocol, outcome definitions, and statistical methods. However, several inconsistencies and omissions should be addressed to align fully with the SAP and registrations and ensure transparency.

General response

We are very grateful to the reviewer for the thorough evaluation and constructive comments. We particularly appreciate the careful scrutiny of our manuscript against the statistical analysis plan and trial registrations. The feedback has been invaluable in helping us to enhance the transparency and rigor of our study. We have addressed each point in detail below, and made corresponding revisions to ensure full alignment and clarity.

My specific comments are as follows:

1. In the 2025/03/04 trial registration update, the PI revised several outcome measures, including the addition of a second primary endpoint: a composite safety outcome comprising death, intracranial hemorrhage (ICH), and malignant brain edema (MBE). Several secondary outcomes were newly specified or clarified, including mRS 0–1 and 0–3 at 90 days, mortality as a standalone endpoint, infarct size by MRI at days 14–21, and ordinal shift in mRS. It is important to note that this revision occurred after the trial had concluded, raising serious concerns about changes to both the primary outcome and analysis methods after data unblinding. The sample size calculation was based on the original primary outcome as a continuous outcome and was not updated to reflect the revised binary outcome(s). The study is likely underpowered for many binary outcomes.

We sincerely thank the reviewer for highlighting the issue regarding the addition and clarification of endpoints. We apologize for the previous oversight in omitting some secondary endpoints from the trial registry. These outcomes were included in the original research protocol as approved by the ethics committees, but were inadvertently not entered into the registry. Prior to peer review, upon clarification with the handling Editor, we updated the registry to include all pre-specified endpoints (i.e., the 2025/03/04 update).

We would also like to clarify that the original primary outcome was indeed binary, and the sample size estimation was based on a heuristic recommendation of 120 participants (60 per group) for binary primary efficacy outcomes (see *reference* below for details). The underlying calculation assumed a hypothesized, clinically meaningful effect size of 0.15–0.25 for the primary efficacy outcome. The sample size was then determined based on simulation studies, which demonstrated that for Phase II trials with binary outcomes, enrolling 60 participants per group (120 total) provides the optimal balance between precision of event rate estimates and minimizing the size of the external pilot trial. This approach is robust across a

range of plausible effect sizes (0.1 to 0.5) and is supported by established statistical methods (please also see *reference* for details). After accounting for an estimated 2.5% rate of suboptimal scan quality, 5% loss to follow-up, and 10% suboptimal recanalization based on the track records of participating centers, a total of 140 participants was targeted.

We deeply apologize for the ambiguity in the original description of the primary efficacy endpoint, which was stated as “*Change of modified Rankin Score to measure degree of disability/dependence. Scores 0-3 is considered good outcome, while scores 4-6 is considered poor outcome.*” This was intended as a binary outcome (good vs. poor outcome). To ensure full transparency, we provide details of the protocol changes in **Supplementary Note 1**. Furthermore, we have only kept the explanation on sample size estimation based on a binary primary outcome, i.e. mRS 0–2 at 90 days, in the **Statistical Analysis Plan** to avoid any ambiguity.

Finally, we fully agree with the reviewer that, given the Phase II design of this trial, the findings should be considered hypothesis-generating and require validation in a Phase III study. As our study’s primary endpoint is the efficacy outcome, we have refrained from reporting *p*-values for other outcomes, including the primary safety outcome, which is instead highlighted descriptively to underscore its clinical importance. All secondary and post-hoc analyses, including the interaction between intravenous thrombolysis and treatment effect of semaglutide, should be interpreted as exploratory. We hope these clarifications fully address the reviewer’s concerns (page 12, paragraphs 1–2): “*Owing to the limited sample size inherited to the phase 2 trial design and the exploratory nature of this analysis, the possibility of type I error cannot be excluded. Thus, an adequately powered phase 3 study is needed to explore whether the potential neuroprotective effect of GLP-1RA is more pronounced in strokes with later presentations. Our study has several limitations. First, the current phase 2 trial was not powered to draw definite conclusions regarding the efficacy of semaglutide in improving neurological outcomes in patients with LVO, but rather to estimate any potential treatment effects and evaluate the safety and tolerability of semaglutide treatment to inform the design for an adequately powered phase 3 trial.*”

Reference

Teare D, Dimairo M, Hayman A, Shephard N, Whitehead A, Walters S. Sample size requirements for pilot randomised controlled trials with binary outcomes: a simulation study. *Trials*. 2013 Nov 29;14(Suppl 1):O21.

2. The IVT subgroup analysis was not pre-specified in the SAP. As a post hoc analysis, it is vulnerable to data-driven inference, including cherry-picking and inflated type I error rates.

We thank the reviewer for raising this important point. In the revised manuscript, we have highlighted the sample size limitation and clearly stated the exploratory nature of the analysis involving the IVT vs. no-IVT stratum. In addition, we have limited the reporting of *p*-values for the comparisons in the IVT subgroup, and emphasize the secondary/exploratory nature of these analyses in the revised manuscript (page 12, paragraph 1): “*Owing to the limited sample size inherited to the phase 2 trial design and the exploratory nature of this analysis, the possibility of type I error cannot be excluded. Thus, an adequately powered phase 3 study is needed to explore whether the potential neuroprotective effect of GLP-1RA is more pronounced*

in strokes with later presentations.” We also further acknowledge that (page 12, paragraph 2): “Our study has several limitations. First, the current phase 2 trial was not powered to draw definite conclusions regarding the efficacy of semaglutide in improving neurological outcomes in patients with LVO, but rather to estimate any potential treatment effects and evaluate the safety and tolerability of semaglutide treatment to inform the design for a large phase 3 trial.”

3. The SAP explicitly states that missing data should be imputed with the worst possible score (e.g., mRS = 6, NIHSS = 42), but this is not described in the manuscript. In addition, baseline characteristics of participants lost to follow-up should be compared with those who completed the study to assess potential bias due to missing data.

We thank the reviewer for the careful examination. We have now more clearly described imputing missing outcome data with the worst possible score in the revised **Methods** (page 18, paragraph 1): *“Missing outcomes were imputed as the worst possible score for an outcome measure, i.e. an mRS of 6 and an NIHSS of 42.”* We have also compared the baseline characteristics of participants lost to follow-up with those who completed the study (**Table S2**).

4. The SAP indicates that an interim safety analysis was to be performed after the enrollment of 70 participants, but this is not mentioned in the manuscript. Reporting the conduct and results of the interim analysis is important for transparency and to confirm that continuation of the trial was not conditional on unreported findings.

We apologize for this oversight. We have reported the results of the interim analysis in the updated manuscript (page 7, paragraph 1): *“A pre-specified interim safety analysis conducted after the first 69 patients completed the study suggested no indication of increased risk of intracranial hemorrhage, malignant brain edema, neurological deterioration (as measured by changes in NIHSS from baseline to day 3), or poor neurological recovery (mRS 4–6 at 90 days) with semaglutide therapy (Table S4).”* Additionally, we wish to emphasize that all adverse events listed in **Table 3** were subject to mandatory reporting to the ethics committees in accordance with hospital regulations.

5. The trial registration dated 2024-09-12 states that blood–brain barrier (BBB) permeability would be assessed via CT perfusion from Day 0 to Day 90. Although the SAP lists this as a secondary imaging outcome, no results or justification for its omission are provided in the manuscript.

Thank you for the comment. Indeed, our initial plan was to evaluate blood-brain barrier (BBB) permeability using K_{trans} derived from CT perfusion imaging. However, during the course of the study, we encountered significant methodological challenges. In cases of large vessel occlusion (LVO), critical hypoperfusion prevents contrast agents from reaching vulnerable brain regions. As a result, K_{trans} has not been validated as a reliable marker of BBB permeability in this specific context. After consultation with our biomedical engineering team, we recognized that additional research is necessary to accurately model the relationship between K_{trans} and BBB permeability during LVO. This includes accounting for delayed contrast arrival times and conducting associative studies with outcomes such as intracranial hemorrhage and malignant brain edema. Furthermore, the low incidence of these events in our

cohort (intracranial hemorrhage: $n = 15$; malignant brain edema: $n = 12$) limits the statistical power for robust validation.

We plan to address these limitations in our subsequent Phase III trial. In particular, we aim to incorporate proteomic markers associated with BBB breakdown, such as matrix metalloproteinase-9 (*reference*), to complement imaging-based assessments. We have acknowledged the limitation in the revised manuscript as elaborated in the response to point 6.

Reference

Kollikowski AM, Pham M, März AG, Feick J, Vogt ML, Xiong Y, Strinitz M, Vollmuth C, Essig F, Neugebauer H, Haeusler KG, Hametner C, Zimmermann L, Stoll G, Schuhmann MK. MMP-9 release into collateral blood vessels before endovascular thrombectomy to assess the risk of major intracerebral haemorrhages and poor outcome for acute ischaemic stroke: a proof-of-concept study. *EBioMedicine*. 2024 May;103:105095.

6. The SAP specifies that transcriptomic and biomarker data were to be collected at multiple timepoints. However, the manuscript only refers to these briefly without providing details or results.

We thank the reviewer for requesting us to clarify on transcriptomic and biomarker data. While we have successfully stored blood samples from study participants for omics analyses, resource limitations prevented us from processing these specimens for transcriptomic and proteomic evaluation during the current study period. We plan to address this gap by pooling these specimens with those collected in our upcoming Phase III trial, to enable a comprehensive assessment of GLP1-RA-induced transcriptomic and proteomic changes in acute ischemic stroke.

To ensure transparency, we have reported the limitations in the revised manuscript (page 12 paragraph 2): *“Fifth, the original plan to assess BBB permeability as a radiological outcome and to conduct omics analyses was not implemented due to the lack of model validation and resource limitations, respectively. Future studies should incorporate BBB leakage-associated biomarkers (e.g., matrix metalloproteinase-9) to more effectively evaluate the impact of GLP-1RA on the BBB.”*

7. The manuscript states that data are available upon request but does not specify what data will be shared, how requests will be evaluated, or whether a data-sharing platform or repository will be used.

We thank the reviewer for this reminder. We have updated the Data Availability Statement (page 19, paragraph 2): *“Anonymized data, including age, sex, admission NIHSS, pre-morbid mRS, treatment allocation, and study outcomes will be made available upon reasonable request to the corresponding authors (Bonaventure Y. Ip, email: ipyiuming@gmail.com, or Ho Ko, email: ho.ko@cuhk.edu.hk) from qualified investigators, beginning 3 months and ending 5 years following publication for academic purposes. Source data are provided with this paper.”* We have also uploaded the source data relevant to the manuscript to figshare with the submission.

8. The manuscript does not describe how fidelity to the intervention (e.g., timing and adherence to semaglutide administration) was monitored or ensured.

Thank you for the comment. The dosage and timing of semaglutide administration were recorded using the computerized inpatient medication order entry system at the participating centers, where barcode scanning of patient bracelets is mandatory during drug administration. In this study, all participants in the semaglutide group received the second injection on day 7, either in the stroke ward or the rehabilitation ward that uses the same computer system within the hospitals, except for three participants who did not receive the second dose due to mortality prior to day 7. We have included this information in the revised **Methods** (page 15, paragraph 1): “*Semaglutide administration was documented using the computerized systems at the participating centers, either in the stroke unit or rehabilitation unit.*”

9. Table 1 should report baseline characteristics rather than outcomes, and it would benefit from reorganization and removal of marginal p-values.

We thank the reviewer for the suggestion. In the revised manuscript, we now present only the baseline characteristics in **Table 1** and have removed the marginal *p*-values. The secondary continuous outcomes are now displayed in **Table 2**.

10. The CONSORT diagram should adhere more closely to established CONSORT reporting guidelines.

Thank you for this advice. We have revised the CONSORT diagram accordingly (**Figure 1**) to more closely align with the CONSORT reporting guideline, detailing the enrolment, intervention allocation, follow-up, and data analysis:

General Responses and Overview of Revision

We are grateful for the comments and constructive criticisms from Reviewer #4, which have helped us greatly when revising the manuscript. Please see the following point-by-point responses to each comment.

Point-by-point Responses

Reviewer #4 (Remarks to the Author):

1. **“1.(Line 52) The primary endpoint should be clearly specified here, identifying which outcome serves as the main endpoint.**

11. (Line 293) The endpoint configuration here is confusing—is safety or efficacy designated as the primary endpoint? This ambiguity must be clarified to ensure methodological rigor.”

I acknowledge the author's detailed explanation regarding the primary endpoint, which clarifies their main intent. However, the description of the safety endpoint remains ambiguous: why is it termed 'primary safety endpoint'? What does 'primary' imply here? Is there prespecified statistical power for this endpoint? Literally, it appears equivalent to the mRS assessment. Moreover, this modification contradicts the protocol description. In the following protocol description, dual primary endpoints are still designated.

“2.5. Endpoint measurement

Primary efficacy outcome is D90 mRS, mRS 0-2 is considered good outcome, while mRS 3-6 is considered poor outcome. Primary safety outcome is a composite of death, MBE and ICH.”

We thank the reviewer for the valuable comment. To avoid ambiguity, we reported the composite safety endpoint as a secondary outcome. We have specified this change in the Methods section (page 16, paragraph 3):

“In the study protocol, a primary composite safety endpoint comprising death, MBE, and ICH was defined. This endpoint was analyzed as a secondary outcome, as no statistical power had been prespecified for its assessment.”

2. **Furthermore, the author proposes a composite primary endpoint, which should be used with caution. The clinical significance of endpoints such as death and malignant edema may not be equivalent, and it is unclear whether they are assessed at a single follow-up time point.**

We thank the reviewer for the comment. The use of a composite safety endpoint in our study was intended to detect any potential harm given the phase 2 study design. However, we fully acknowledge that component events differ in their clinical significance which necessitates careful interpretation. We have also presented the results for individual components as secondary endpoints, reporting death at 90 days, malignant brain edema and intracranial hemorrhage (page 8, paragraph 1).

Regarding assessment timing, all patients underwent plain CT brain imaging between 4 and 7 days post-randomization or during periods of clinical deterioration to identify

intracranial hemorrhage or malignant brain edema. Death at 90 days were evaluated via clinical follow-up or telephone/video conferencing at 90 days after randomization (page 15, paragraph 2).